



# Floating Offshore Wind in Japan: Addressing the Challenges, Efforts, and Research gaps

Ryota Wada[1], Amir R. Nejad[2], Kazuhiro Iijima[3], Junji Shimazaki[4], Mihaela Ibrion[2], Shinnosuke Wanaka[5], Hideo Nomura[6], Yoshitaka Mizushima[1], Takuya Nakashima[1], and Ken Takagi[1]

[1]Graduate School of Frontier Sciences, The University of Tokyo, Japan
[2]Department of Marine Technology, Norwegian University of Science and Technology, Trondheim, Norway
[3]Graduate School of Engineering, University of Osaka, Japan
[4]Equinor, Tokyo, Japan
[5]National Maritime Research Institute, Tokyo, Japan
[6]Kiso-Jiban Consultants Co., Ltd., Tokyo, Japan

**Correspondence:** Amir R. Nejad (Amir.Nejad@ntnu.no)

**Abstract.** This paper aims to identify the technology gaps on the path to achieve sustainable development of floating off-shore wind in Japan. Japan has a strong desire for floating offshore wind development motivated by energy security, climate change, and industry promotion. The key challenges are described with an emphasis on the unique environmental conditions of Japan, such as earthquakes and tropical cyclones. In addition, the absence of oil & gas development in Japan has led to social

challenges such as a lack of supply chain, infrastructure, and human resources in offshore wind. A review of state-of-the-art technologies is provided in each technology domain for four research domains: site selection, design & operation, industry development, and contribution to national energy policy "S + 3E" or "Safety + Energy security + Economic efficiency + Environmental sustainability". The gaps identified in the paper suggest the need for specific research topics such as the assessment of unique environmental conditions, the design of robust and cost-effective structures considering fabrication, transportation,

installation, and operation, and the data sharing strategy for efficient and rapid learning. In addition, many challenges show technology gaps across domains, indicating the need for interdisciplinary research collaboration and system integration of complex systems with digital engineering approaches. Finally, the need for the development of an industry roadmap is discussed.

## 1 Introduction

### 1.1 Motivation

Changes in climate indicators, such as temperature, precipitation, sea level rise, ocean acidification, and extreme weather conditions, have forced us to be aware of the severity of climate change impact on natural and human systems. At the same time, the rapid growth of the world population raises concerns about the planet's ability to sustain life for humans and other species. Climate change and sustainability has become one of the biggest concerns of the international community.

The vast ocean that covers 70% of the Earth's surface plays a crucial role in modulating our climate system by regulating the

cycle of heat, water, carbon, and other elements. Humans have historically relied on ocean resources for fishing, transportation,





and also oil & gas resources since the last century. Over the last decades, ocean industries have shifted to the interdisciplinary ocean industries or ocean based industries which are covering energy production such as offshore wind, wave tidal, floating solar, food production at sea, deep sea mining, transport in harsh environmental conditions of Arctic and Antarctic regions, just to mention a few of them. All of these ocean-related industries which all contribute to the blue economy are pushing the
borders deeper and further offshore. (Nejad, Amir R. and Ibrion, M., 2021)

Under the strong tailwind of decarbonization, the offshore wind industry has evolved significantly driven by the high interest both in the scientific and industrial arena. The installed offshore wind capacity of the world increased from 3.1 GW in 2010 to 35 GW in 2020, and many upcoming projects are distributed globally. Many countries have set ambitious targets for offshore wind, e.g., the European Union aims at 110 GW by 2030 and 400 GW by 2050. The capacity of offshore wind turbines has
become bigger and larger, such as a 16MW rated turbine with a rotor diameter of 236 m (Mehta et al., 2024).

Current offshore wind projects are installed mainly as bottom-fixed structures, such as monopile, gravity-based, and jacket structures, that operate in shallow water up to 50-60m depth. Floating substructures allow wind turbines to operate in deeper waters, providing access to a broader ocean area and larger potential offshore wind energy resources. Compared to bottom-fixed structures, the industry is still in its infancy. However, global capacity of floating offshore wind turbines (FOWTs) is
predicted to grow to more than 200 GW by 2030, for example (IRENA, 2019).

Japan also has a strong interest in offshore wind energy. The island nation faces numerous energy-related challenges, including a heavy dependence on imported fossil fuels, nuclear power concerns following the Fukushima disaster in 2011, and the need to reduce greenhouse gas emissions to fight climate change. Concerns about energy security, decarbonization, and supply stability have pushed Japan to explore offshore renewable energy, particularly offshore wind (Ibrion et al., 2020b).

The topography around Japan's coast has a steep slope, unlike the stretch of shallow water seen in Europe. In addition, the coastal area has various activities including fishing and ship routes, which sets a limitation in the introduction of bottom-fixed offshore wind energy in Japan. Floating offshore wind can access the world's 6th largest Exclusive Economic Zone (EEZ) of Japan. Considering the target of up to 45 GW of offshore wind power by 2040, massively deploying floating offshore wind turbines in deep water areas, including EEZs, is inevitable. A report estimated the potential of offshore wind in Japan as 116
GW for bottom-fixed and 2940 GW for floating offshore wind, including the EEZ (Institute, 2024).

The Public-Private Council on Enhancement of Industrial Competitiveness for Offshore Wind Power Generation set an ambitious target for Japan's offshore wind energy (on Enhancement of Industrial Competitiveness for Offshore Wind Power Generation, 2020): 10 GW by 2030, 30-45 GW by 2040 (including floating offshore wind), domestic procurement ratio: 60% by 2040, and Cost reduction: 8-9 yen/kWh by 2030-2035 (bottom-fixed). These targets were prepared jointly by industry and
government as it plays a critical role for both parties to build a sustainable ecosystem around offshore wind. The government needs to designate sites and introduce legislation accordingly. The predictability of future market opportunities is expected to attract investment critical for fostering a cost-competitive and scalable energy source. In addition, Japan plans to announce targets for floating offshore wind in the near future.





## 1.2 Knowledge Gap addressed in the paper

Japan's push for offshore wind energy reflects its broader goals of achieving energy security and sustainability. However, successful offshore wind projects must be attractive to investors. For floating offshore wind turbines to be accepted as one of the main power sources in Japan, the LCOE (levelized cost of energy) must be reduced to an order of 10 JPY/kWh (about 0.11 USD at the exchange rate of 2021) or lower to achieve grid parity. In the first offshore wind auction in Japan in 2021, FIT (Feed-in Tariff) for a floating offshore wind project at offshore Goto was 36 JPY/kWh. That is a much higher FIT than the

other bottom-fixed projects in the same auction round, from 11.99 JPY/kWh (0.13 USD) to 16.49 JPY/kWh (0.18 USD). A bold cost reduction will be necessary.

Another important factor for offshore wind projects to be economically feasible is the project to be bankable. The large capital investment required for offshore wind projects are financed with either corporate finance or project finance. In either case, offshore wind projects often cover 70-80% of the capital with debt (Aono, 2018). Many owners and financiers have gained

experience in the European bottom-fixed offshore wind market. They are aware of potential market risks, competition needs among suppliers and developers, as well as economic feasibility (Azevedo and Grosse, 2019). In the case of Japan, it is critical to create a foreseeable market by showing profitability and stability. In addition, scalability is also necessary to contribute to energy security. These projects must be economically feasible, bankable, and scalable.

In general, floating offshore wind is still a nascent technology and many concepts are under development. Some promising

concepts have scaled up and entered the commercial phases. Here, we provide a general view of how technology development, business development, and policy can contribute to overcome the key barriers.

Technology development is a key driver for cost reduction. Innovative designs such as larger turbines with higher capacity, floating offshore wind turbines, and improved materials can lead to more efficient and less expensive energy generation. Developing technology to simplify and reduce the cost of installation and maintenance is crucial. This includes improvements in

vessel design for installation and maintenance, as well as enhanced forecasting and monitoring systems.

Business development to demonstrate the long-term profitability and stability of offshore wind projects is important. Innovative financing mechanisms, such as green bonds or public-private partnerships, can be useful. Developing a robust supply chain and encouraging the growth of the local industry can reduce costs and support the feasibility of the project. This includes building local manufacturing facilities for components and creating jobs, which can also improve public support for offshore

wind projects. Planning infrastructure, such as port and grid, for large-scale deployment and standardization of project components and processes can help scaling up the nation-wide capacity. As larger floating offshore wind projects start to come online, economies of scale is expected to lower the manufacturing and fabrication cost. In addition, as the size of each turbine grows in capacity, there will be fewer platforms in commercial-scale wind farms of the same capacity. The competitive nature of the bidding process, along with the industry's learning curve, is also expected to help drive down costs.

Finally, the role of policy, both regulatory and promotional, in supporting offshore wind is vital. Streamlined permitting processes, clear environmental guidelines, and consistent policies will provide long-term stability for investors. Ambitionate but achievable long-term goals committed by the government can drive aligned investments throughout the industry. In addition,





incentives such as feed-in tariffs, tax credits, or subsidies can make projects financially more attractive by offset the high initial capital costs and improve overall economics.

Despite the many lessons learned from the experience in bottom-fixed offshore wind projects, there are still many barriers we need to overcome for the sustainable development of floating offshore wind. In addition, the environmental, social, and technological conditions surrounding the offshore wind industry are unique to each country. Differences in system requirements, design rules, and cost effectiveness can pose new risks and challenges to development. For example, tropical cyclones (called *typhoons* in this region) and major earthquakes are obvious risks for offshore activities in Japan.

At the same time, some unique aspects can provide opportunities. Although Japan has less experience building large offshore platforms for oil & gas industry, but has a competitive shipbuilding industry with highly productive shipyards. The automotive industry in Japan is known for its efficient and lean supply chain.

Thus, the strategy and path for sustainable development need to be customized considering the circumstances, with regard to the *problem space* and the *solution space* for the floating offshore system. Effective investments in R&D of floating offshore
projects are necessary to adequately address the challenges and realize the opportunities.

## 1.3 Aim of paper

A sustainable development pathway that mitigates risks and maximizes opportunities is desired. The first step will be to identify the environmental, social, and technological conditions surrounding the floating offshore wind industry in Japan. However, the knowledge around floating offshore wind is dispersed in government, industry, and academia, creating a large knowledge gap
to understand Japan's unique situation.

The purpose of this paper is to identify the technology gaps on the path to achieving sustainable floating offshore wind in Japan. Such findings are important for accelerating and prioritizing future research activities around floating offshore wind in Japan. Following the aim of the paper, the research questions in this paper are defined as follows.

**Research Questions**

– What are the unique challenges surrounding floating offshore wind in Japan?

– What research effort is currently being conducted to address these challenges?

– What are the gaps and what shall be prioritized?

## 1.4 Overview and Structure of paper

This paper provides a review of the challenges and efforts surrounding the development of floating offshore wind in Japan.
This research is uniquely organized by industry and academia from across disciplines around floating offshore wind with collaborative efforts between Japan and Norway, covering the practice in other countries, with an emphasis on Norway.

The paper is organized as follows. In the following sections 2, 3, 4, 5, we review the unique situation in Japan to identify challenges to sustainable development and provide a review of state-of-the-art research in technology fields to identify research





gaps. Given many ways to structure the paper, we organize our discussion as follows. The site selection section covers the
unique environmental and social conditions in Japan. Design & operation focuses on the technology development around
FOWTs. Industry development discusses the scalability of FOWTs in Japan. The term S + 3E, which means safety, energy
security, economic efficiency, and environment, discusses the expected role of FOWTs in Japan. Finally, a recommendation for
future studies synthesizing the findings of each section is provided in section 6.

## 2  Site selection, and project planning

### 2.1  Overview

Site selection, planning, and project development are crucial stages in establishing an offshore wind project that is viable,
compliant with regulations, highly effective in energy production, and minimal in environmental impact. Site selection in-
volves evaluating wind resources, environmental impacts, sea bed conditions, proximity to shore, connection to the grid, and
regulatory requirements to choose the optimal location. Planning includes designing the layout, conducting environmental
assessments, obtaining permits, financial planning, and risk management. A thorough investigation of the site is conducted,
including archaeological, geological, morphological, and metocean studies.

The geological site investigation is covered by mainly two categories, the geophysical and the geotechnical survey. The
geophysical survey consists of the seismic surveys such as multibeam echo-sounder, side scan sonar, subbottom profiler, high
resolution streamers, and magnetometer survey for unexploded ordnances. The geotechnical survey includes the standard pen-
etration tests, the cone penetration test and the soil sampling by the offshore borehole and drilling (Randolph and Gourvenec,
2017). With the result obtained, the technical feasibility and the associated cost are evaluated. The risk of encountering un-
exploded ordnances (UXOs) is also considered (Malhotra and O'Connell, 2024). These investigations are often time and cost
consuming, but high-quality site surveys help reduce risk and ensure optimal design.

### 2.2  Site evaluation

The development of floating offshore wind farms goes through a series of processes by the government and industry, follow-
ing the "Act on Promoting the Utilization of Sea Areas for the Development of Marine Renewable Energy Power Generation
Facilities" enacted in 2018. The guideline to designate the offshore field to the project zone were issued by METI (Ministry
of Economy, Trade and Industry) and MLIT(Ministry of Land, Infrastructure, Transport and Tourism). In the guideline, three
categories of such zones are set in the guideline as Preparatory zone - initial step, Promising zone, second step, and Promo-
tion zone, last step before the public auction(METI and MLIT, 2021). Here, prefectural municipalities play the key role in
determining the zone by informing the field situation and their intention to the central government to promote the offshore
wind project in their field. Nevertheless, various developers initiate their development works before the zone designation by
the government, to be the initial and leading developer of the field. Examples of the developer's effort during the early project,
in addition to estimating the project's economic feasibility, are grid investigation, stakeholder identification, and project zoning





support. Detail description of each phase and their challenges are described in the appendix. The main challenges are lack of efficiency in the streamline surveys, Marine Spatial Planning (MSP) for effective management, and need for data sharing and accurate marine spatial data.

Regarding learning from the Norwegian experiences, the Hywind Tampen is the first floating offshore wind farm on the Norwegian Continental Shelf (NCS). This farm will not provide electricity to the Norwegian grid, but it will cover approximately 155 35% of the electricity needs of the five oil and gas platforms located in the Snorre and Gullfaks fields to decarbonize the oil and gas industry. The Hywind Tampen benefited from authorization granted by the Ministry of Petroleum and Energy based on the previous licenses for survey, exploratory drilling, and petroleum activities, and obtained an authorization from the Norwegian Petroleum Act as was seen as a change to the plans for development and operation and as a modification done to the power supply of the oil and gas platforms. They did not require an authorization from Norwegian *Havenergylova* or *Act on renewable* 160 *energy production at sea*, which would require much longer time and procedures to be in place. The Hywind Tampen project is considered to be among the key particularities of the Norwegian road map and its offshore wind market (Ibrion and Nejad, 2023; Herrera Anchustegui, I., 2020; Herrera Anchustegui, 2020).

## 2.3 Licensing

Following site selection, a bidding process is held to select the operator to develop the project. These operators are granted 165 permission to occupy the promotion area for a maximum of 30 years. Six criteria are set to be the promotion zone that include i) appropriate metocean conditions and power generation, ii) no conflict with existing infrastructures such as port and shipping route, iii) available ports for the construction, operation and maintenance of the offshore wind farm, iv) grid connection, v) no adverse effect projected on the fishing industry, and vi) no overlap in the area of fishing ports, public ports, and preserved beach.

The first bidding round was held in three areas in 2021. It surprisingly faced criticism for having too much emphasis on the tender price. Based on the claims, the government revised the evaluation standards for the second round to promote earlier operational commencements and to put limits on the operator's total capacity to 1 GW per auction. The future bidding process is expected to align with the government target, including floating offshore wind. These licensing processes are taking a more cautious approach compared to other countries. The slow speed and limited scale of projects can be a disadvantage with greater 175 risk and less cost control.

Developers make various efforts to lead the offshore wind farm development. This competition causes repeated communication with local stakeholders, site investigation, grid examination, environmental impact assessment, negotiation with the land owner, and consultation with port authorities. In some cases, more than five developers plan the site investigation in the same field. These duplications burden local stakeholders and municipalities, and valuable resources are not utilized effectively 180 among stakeholders and developers. A more centralized public auction system is important for the efficient development of FOWTs.



## 2.4 Regulation

### 2.4.1 Technical standards

Offshore wind power generation, both bottom fixed and floating, facilities, and maintenance methods must comply with the technical regulations specified by the METI and MLIT ordinances. These technical regulations are clear, but general and abstract. Industrial standards are used for complements. Japanese Industrial Standards (JIS) are established in accordance with ISO (International Organization for Standardization) and IEC (International Electrotechnical Commission) standards for their application in Japan. However, due to translation into Japanese and the addition of supplementary explanations, new or updated JIS standards are not established or updated until several years after the ISO/IEC standards are changed. On the other hand, an almost automatic procedure ensures rapid implementation in Europe, and a broad participation of different stakeholders ensures support. In addition, unique concepts and calculation methods for buildings and support structures (e.g., guidelines published by the Japan Society of Civil Engineers, JSCE) are conventionally used in Japan. The differences from international standards are mainly related to the differences in meteorological and geological conditions in Japan compared to Europe: earthquakes, seabed, wind (tropical cyclones) (European External Action Service (EEAS), 2022). There are several differences between the design process of the offshore wind power generation facility in Japan and Europe due to the above challenges, and these differences result in Japan's specific design requirements.

### 2.4.2 Environmental assessment

In accordance with the Environmental Impact Assessment Act (Act No. 81 of 1997), the environmental impact assessment is carried out before business activities, and the results of the assessment are open to the public to obtain opinions from the national government, local governments and citizens, and these opinions are taken into account from the point of view of environmental conservation. Special provisions of the Electricity Business Act are also applied for offshore wind projects.

In 2019, the Renewable Energy Sea Area Utilization Law, established by METI and MLIT, designated promotion areas within its territorial waters where offshore wind projects can be implemented, selected companies through auctions and allowed long-term occupancy (for a maximum of 30 years). The Renewable Energy Sea Area Utilization Law and the Environmental Impact Assessment Law are independent, and the existing environmental assessment system is applied in parallel. The selected companies are required to conduct an environmental assessment based on the law separately. However, in Denmark, the Netherlands, etc., the national government takes the lead in conducting environmental assessments to a certain extent.

In 2021, the Ministry of Environment (MoE) established Regional Decarbonization Roadmap (June 2021) and Global Warming Prevention Plan (October 2021) for promotion of wind power generation through optimization of environmental assessment focused on offshore wind. Environmental assessment system for offshore wind is to be optimized in cooperation with relevant ministries and agencies in Japanese national government, local governments, and business operators. In addition, to promote offshore wind, the methodology of environmental protection is to be considered referring to overseas experiences. The Regulatory Reform Implementation Plan (approved by the Cabinet in June 2022) was established to optimize the environmental assessment system considering the characteristics of offshore wind power generation, such as location and environmental





impact, in cooperation with relevant ministries, local governments and developers. MoE organized a committee to discuss optimization of the environmental impact assessment system for offshore wind from May to July 2023 and defined the role of government, to correct a wide range of information and knowledge from local stakeholders from the early stage before selecting an offshore wind developer and reflect the results of the surveys in the area selection process. Through continuous condition monitoring during the construction and operation phases, overall environmental impact of the entire country from
offshore wind development can be reduced. The government will promptly establish new environmental assessment system, including consideration of necessary legislation, based on the results of the committee.

## 2.5 Geology and Geohazards

The features of seafloor topography differ greatly between passive margin, such as continental margins in Europe, and active margin, such as island and land arcs in Japan. In Europe, offshore wind farms are located mainly in areas of the North Sea,
which is shallower than 200 m deep (light areas on the right side of Fig. 1. These areas have shallow water and little fluctuation in seabed topography. In contrast, the area in the waters around Japan where the depth of the water is less than 200 m (light-colored areas on the left side of Fig. 1), which is the target of offshore wind power development, is very limited compared to the North Sea. In addition, floating offshore wind farms that are expected to be developed in the EEZ of Japan will have to be developed in areas where the topography of the seafloor is complex and steep. In areas with complex topography, such as the
Japan Sea off the Tohoku region, sedimentation rates are expected to have spatial variance (Katayama and Itaki, 2007). Thus, the sea floor where anchors will be installed may consist of a variety of geological compositions, such as very soft sedimentary layers consisting mainly of clay, sedimentary layers consisting mainly of sand and gravel, soft bedrock, and hard bedrock. These differences in the geology comprising the seafloor will affect the selection of anchor type. If the seafloor consists of bedrock, drag anchors or suction anchors, which are used in many floating offshore wind turbines, cannot be employed. In this
case, gravity anchors or pile anchors are required. If the bedrock is very hard, gravity anchors are necessary, as it is difficult to drive the piles. If the seafloor consists of very soft cohesive soil, suction anchors cannot be used as the required pull-out resistance cannot be ensured, drag anchors and gravity anchors must be submerged to a very deep depth in the ground to obtain the necessary holding force. In the case of pile anchors, it is also necessary to drive very long piles into the ground to obtain the required friction force on the skin. The geological composition of the seafloor has a great influence on the selection and
design of anchor types; therefore, it is necessary to conduct geological investigations at numerous locations in a subject area to understand seafloor conditions in detail.

Geological investigations for offshore wind power development can be divided into two types: investigations conducted by drilling from a Self-Elevating Platform (SEP) which is installed offshore and investigations from a vessel or using an equipment lowered from a vessel on the seafloor. The former is called "SPT investigation" as Standard Penetration Test (SPT) is usually
conducted, while the latter is called "CPT investigation" as a Cone Penetration Test (CPT) is usually performed. During the SPT investigations a sampler is driven into the ground by hammer blows and the number of blows is measured, along with collecting the undisturbed soil samples for laboratory tests, and various in-situ tests using drill hole. The evaluation system for ground properties in Japan has a history of being built mainly on this SPT investigation and is considered to have some





reliability as a ground evaluation method. Meanwhile, during the CPT investigation a cone equipped with sensors is penetrated into the ground to estimate the ground properties. It is also possible to conduct soil tests on disturbed soil samples taken during the CPT investigation. Compared to SPT investigations, CPT investigations require less time and are thus suitable for investigation at a large number of locations in a short period of time. Since CPT investigations had been rarely conducted in Japan, their reliability as a ground evaluation method is considered unclear. Therefore, it is common to combine SPT and CPT investigations in order to improve the reliability of CPT investigations. In addition, it is difficult to conduct SPT investigations in deep water due to the difficulty of installing a SEP in deep water. However, CPT investigations are relatively easy to conduct even in deep water as CPT investigations are conducted by a vessel. For this reason, CPT investigations are the main type of investigation for floating offshore wind turbines installed in deep water.

Several geohazards exist on the seafloor, as shown in Fig. 2. The effects of geohazards on floating offshore wind include, for example, displacement of anchors, breakage of mooring cables, and breakage of power cables in the event of a seabed landslide or turbidity current. In addition, if liquefaction of the seabed occurs due to an earthquake, the anchors will be affected. The liquefied ground behaves like a fluid, reducing bearing capacity, friction, and counter-forces. Therefore, drag anchors, gravity anchors, and chains sink into the ground by their own weight, changing the formation of the mooring arrangement. In the case of suction anchors, the anchor body can tilt or collapse due to loss of counterforce caused by liquefaction of the surrounding ground. In addition, the suction force may decrease due to an increase in the pore water pressure of the soil inside the anchor, resulting in a loss of hold force. In the case of pile anchors, liquefaction of the soil around the pile may reduce the skin friction force around the pile, which may reduce the pull-out resistance, and consequently the pile anchors may be pulled out of the ground (i.e. displace vertically upward). It is important to evaluate the potential change in soil characteristics after liquefaction occurs and include in design conditions. Geohazards on the seafloor are characterized by their extremely large scale. For example, the distance of flow of turbidity currents, which are often described in contrast to debris flows on land, ranges from tens to thousands of kilometers for large ones. In addition, most submarine landslides are up to $10 \text{ km}^2$ in size in the coastal areas of Japan, and are usually 10 to 100 times larger than landslides on land. Geohazards on the seafloor are not easy to deal with, and the reason for this lies in their large scale and the difficulty of construction on the seafloor. Therefore, in order to install offshore wind power generation facilities, it is necessary to avoid geological risks by properly arranging wind turbines and risk mitigation measures. In some cases, depending on the type and size of the geohazard, development may have to be abandoned.

The National Institute of Advanced Industrial Science and Technology (AIST) has compiled and published a "Marine Geological Map" of the results of its research on seafloor geohazards to date. This is a compilation of the interpreted results of seismic and acoustic surveys conducted by AIST, which provide a broad overview of faults, submarine landslides, underwater debris flow deposits, sediment waves, etc.

Although the "Marine Geological Map" is a useful reference for offshore wind power development, it is the result of a wide-area survey and therefore detailed surveys need to be conducted again for localized development. This means that the influence of seafloor geohazards on offshore wind power development has not been fully recognized. Furthermore, no studies have been conducted considering seafloor geohazards at each stage of the desk study: preliminary study; detailed study. In addition, there



are no guidelines or manuals that show how to systematically survey and assess risks due to submarine geohazards specific to
Japan, and research and human resource development for this purpose have not progressed. Therefore, understanding geological
risks due to geohazards of the sea floor, establishing guidelines, promoting research, and developing human resources are urgent
issues. In addition, the applicable anchor concept must be carefully evaluated based on the evaluated geological conditions and
geohazard risks.

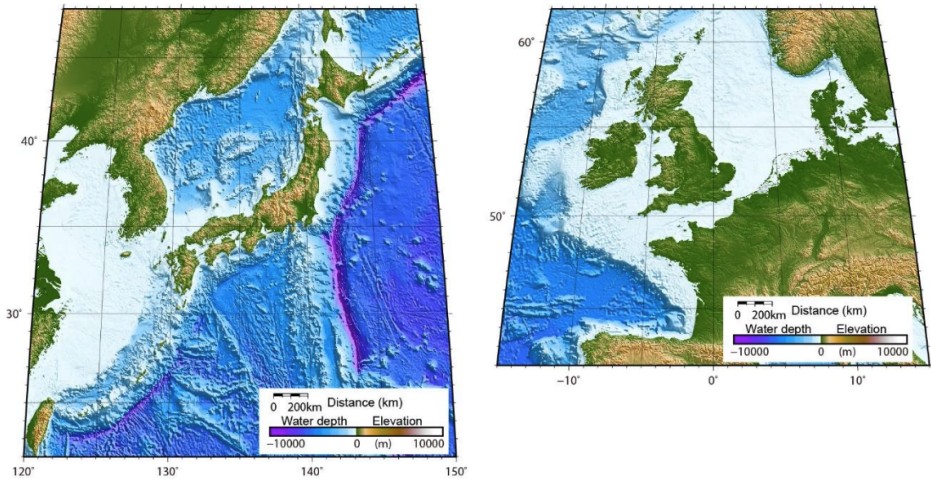

**Figure 1.** Seafloor topography around the Japanese Islands and in the North Sea (both figures are drawn at the same scale (Kawamura, 2023)

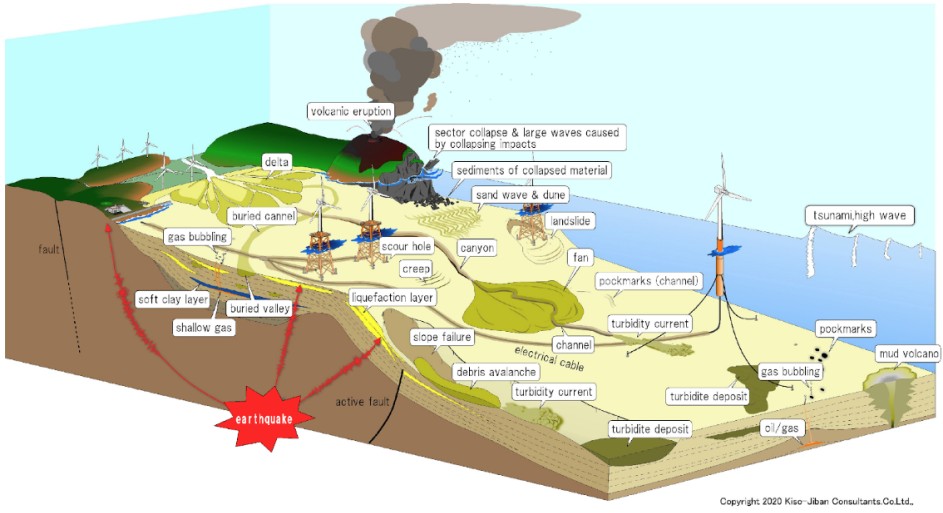

**Figure 2.** Relationship between offshore wind power facilities and seafloor geohazards





## 2.6 Metocean conditions

Metocean is term combining meteorological ("met") and oceanographic ("ocean"), describing comprehensive environment of the atmospheric and oceanic conditions of a particular location. For the development of offshore wind projects, typical metocean data requirements are wind speed, wind direction, air temperature, humidity, and atmospheric pressure for meteorological data, and wave height, wave period, wave direction, sea surface temperature, currents, water depth, and tides for oceanographic data. These metocean data are vital throughout the lifecycle of an offshore wind project. Wind data is used for site selection to

ensure that the site has consistent and strong wind resources. Accurate metocean data is essential for predicting the potential energy yield of an offshore wind farm, providing key information for financial modeling and investment decisions. At the same time, it is used to understand the environmental forces, both normal and extreme conditions, required for the design and engineering of wind turbine structures, foundations, and support systems. Weather forecasts help plan safe and efficient offshore activities.

### 2.6.1 Wind data


Numerical simulations are often used for the initial stage of the wind resource assessment. In 2017, NeoWins (NEDO Offshore Wind Information System) was published to provide offshore wind information necessary to plan offshore wind power generation. The database provides wind statistics with 500m spatial resolution from the WRF simulation. Related information on water depth, submarine geology, and social environment, such as port areas and shipping routes, has been integrated into

the same database to allow easy access. With the data NeoWins provides, the distribution of the average wind speed can be estimated by fitting the Weibull distribution for both the wind farm design and the estimation of the annual energy production during the initial planning phase. Regarding the estimation of extreme wind speed, stochastic simulations such as MASCOT Offshore (Yamaguchi and Ishihara, 2010; Ishihara and Yamaguchi, 2015) are widely used in Japan. In addition, various private companies provide wind condition analysis utilizing publicly available meteological reanalysis data, e.g., ERA5 (Setchell,

2020) from the ECMWF and JRA(Agency, 2013) from the Japan Metrological Agency.

In the later stages, field observations are conducted to i) validate the simulation result, ii) set the design and financial basis (bankability) for the wind farm and iii) prepare for the wind farm certificate and the construction permit. Offshore Wind Measurement Guidebook provides guidance of scanning LiDAR (Light Detection And Ranging) usage (NEDO, 2023; Ueda et al., 2022). These observation are often made with Wind LiDARs placed on the coast for remote observation. For floating

offshore wind projects, wind measurement by Floating LiDAR System (FLS) is required. Here, a single point observation can represent the wind conditions within 10 km of the observation point. Though the measurement of mean wind speed and direction with FLS are accepted, that of the turbulence intensity is still under investigation due to the motion-induced measurement error(NEDO, 2023).




### 2.6.2 Wave data

Initial stage assessment of wave conditions are also made from numerical simulations. Wave simulation is conducted with wave models, such as third generation wave model Wave Watch III and WAM (Suzuki et al., 2016), using the wind data as the input. These models have been developed to estimate the waves at deep water locations, and do not consider shallow water effect(Mase et al., 2017). As the topography around Japan is complex and the seabed is steep, shallow water transformation is needed at the offshore wind site. For feasibility studies of the project development, TodaiWW3 provides high-resolution

wave simulation around Japan(Waseda et al., 2016). Regarding the shallow water transformation, three models are mainly utilized such as EG Wave(Mase et al., 2001), SWAN(swa), and NOWT-PARI(Par)(Hirayama, 2013). Although the shallow water transformation requires water depth information, the limited data of seabed topography is available around Japan. When the water depth data is available from the site geophysical survey, the survey data can be used; otherwise, the existing available data, e.g. GEBCO and M7000 are referred.

About the wave, Nationwide Ocean Wave information network for Ports and HarbourS (NOWPHAS(now)) data is used as the site data if it is available around the project site. Where the NOWPHAS data is not available near the project site, the site measurement is required in order to estimate the hydrodynamic load to the FOWT structure(ClassNK, 2021b).

### 2.6.3 Unique situation and state-of-art research

The development of floating offshore wind power generation in Japan involves challenges that are specific to Japan.

First is the moderate wind condition. A study estimated the realistic wind farm capacity factors by analyzing the wind speeds for each grid location. In comparison with European countries, such as Netherlands and Norway, that show capacity factor above 55% in some regions, the highest value in Japan was around 47% (Bosch et al., 2019). The capacity factor is proportional to the yield of wind farm's power generation and has direct impact on the LCOE.

Second is the extreme sea states in the sea around Japan, which are mainly dominated by tropical cyclones and extratropical

cyclones. These are relatively low-frequency events and the region being directly hit is rare. However, the high wind speed and high waves caused by these cyclones can cause critical damage to the structures. The design of FOWT needs to consider robustness against these extreme events. IEC introduced Class T turbines designed for the high wind speed and turbulence caused by tropical cyclones (Lundquist et al., 2023). For example, the International standard IEC 61400-3 requires the consideration of several design load cases under 50-year extreme storm conditions. These load cases are defined from joint distributions of

wind speed, significant wave height, and the wave period. However, only a small number of samples are available because of the rareness of these events. The large uncertainty in estimating the extreme condition in tropical cyclone-dominated regions is suggested (Wada and Waseda, 2020). Uncertainty quantification (Wada et al., 2016) and measures for uncertainty reduction, e.g. (Wada et al., 2018; Sando et al., 2024), of extreme value analysis are essential for robust and efficient design.

The coast facing the Pacific Ocean is known for strong swells with long wave periods (Snodgrass et al., 1966). Several

cases of port facility damage have been reported for wave heights lower than the design wave height. This may be caused by negligence in load conditions caused by swells, as the design wave heights for extreme conditions are often dominated





by wind waves with shorter wave periods (Matsufuji et al., 2017). The same can be said for offshore operations, where the swell condition exceeds the operational limit of the wave period (typically around 7 seconds) leading to increase of weather downtime.

One of the major disadvantages of Japan's offshore wind industry is the lack of accumulation of metocean data. Regions such as the North Sea and Gulf of Mexico have a long record of metocean data due to the prosperity of the offshore oil & gas industry. Since there are only very limited offshore oil & gas activities near Japan, the amount of available metocean data is limited in quality and quantity. Together with enhanced observation networks, it is important to develop a shared platform for site evaluation data, such as DHI metocean data.

## 2.7 Japanese Central Method

Given the social, environmental, and technical challenges outlined in this section, the Ministries aim to introduce the Central Method. The outline of the Central Method was published in 2023(Ministry of Economy et al., 2024), and its operating policy was issued in 2024(Ministry of Economy and , METI) by METI and MLIT. The operating policy identifies five key areas: i) Designation of offshore wind project areas and public auctions for developers, ii) Regional coordination for project initiation,
iii) Site investigations (wind, soil, topography and metocean), iv) Security of grid connections and v) Environmental considerations. Regarding (i), the policy emphasizes that the government manages offshore wind auctions in accordance with the area designation guidelines and the public auction operating policy. Concerning (ii), the policy commits to supporting local municipalities in the formation of projects by calling for the application of feasibility studies and investigations. However, fisheries-related matters are delegated to local municipalities and developers. For (iii), the Japanese public independent body
JOGMEC has been assigned as the executor of site investigations. ANRE published the basic specifications for site investigations(Ministry of Economy and , METI), stating that the project-specific specifications are prepared individually by JOGMEC. With respect to (iv), the government explicitly intervenes in the grid security scheme to prevent duplicated grid applications and infrastructure construction. The discussion continues on the scheme design. Finally, for (v), the policy notes that the MoE is conducting studies on the Environmental Impact Assessment (EIA) Act as it relates to offshore wind development. The MoE
organizes review committees to discuss the appropriate application of the EIA Act for offshore wind projects.

### 2.8 Summary and key takeaways

Several implications are made on how licensing, regulation, and site evaluation can be improved. First, ensuring transparency in regulatory processes related to offshore wind energy development is essential. Providing clear guidelines, accessible information, and opportunities for public input can help build trust and confidence in the regulatory framework. We need to develop
a comprehensive and unified standard for project development, i.e. applying the same standards, regulations, processes, and calculation methodologies for all projects that can be the single reference document for all offshore wind related regulation. In addition, as there are many unknowns in this immature industry, shifting the principles of regulation from rule-based assessment to risk-based assessment shall be beneficial to balance accountability and cost drivers. Next, the role of government in site evaluation, especially stakeholder management, is essential. Currently, there is a large overlap in the investigation process,





putting a burden not only on developers, but also on local stakeholders and municipalities. To maximize valuable resources among stakeholders and developers, a more centralized public auction system and the promotion of MSP are essential. Finally, a deep understanding of the geological and metocean conditions at the site is necessary. The challenges of considering Japan's unique geological and metocean environment require further understanding of the impact of earthquakes and tropical cyclones. Enhancing the data sharing platform for more efficient site evaluation is important.

## 3   Design & Operation of FOWT

### 3.1   Design of FOWT

The design of FOWT has many figures of merit, such as cost reduction (improvement of the capacity factor, reduction of operating costs, etc.), efficient mass production, and easier maintenance. To reduce project costs of floating offshore wind, NEDO (New Energy and Industrial Technology Development Organization) is conducting the project "Cost Reductions for
Offshore Wind Power Generation" from 2021. The aim of this project is to establish technology that can achieve a power generation cost of 8 to 9 yen/kWh with fixed-bottom wind turbines under certain conditions, and technology to commercialize floating offshore wind turbines at internationally competitive cost levels.

When discussing offshore wind in Japan, it should be noted that there are meteorological, meto-oceanographic, and geographical issues specific to Japan or Asia. The Japanese archipelago is located at latitudes between 20 and 46 degrees north,
and thus hardly benefits from westerlies. However, Japan is in the path of tropical cyclones, and maximum design wind speeds for bridges, for example, can reach 75 m/s or even 100 m/s. In addition, Japan is like an island floating suddenly in the Pacific Ocean at an average depth of 4,000 m. The continental slope is steep, and the shallow water and continental shelf area is small, easily reaching a depth of 1,000 m. The average water depth even on the the Sea of Japan side is about 1600 m. The design of floaters shall take into account such complex and unique conditions.

### 3.1.1   Design of Wind Turbines

The size of offshore wind turbines is increasing and this requires attention to be paid to the component flexibility and potential dynamic coupling effects during design modeling and analysis (Nejad et al., 2022).

Considering the low capacity factor around Japan, collaboration with wind turbine manufacturers to design wind turbines and floating foundations that are suitable for the Japanese and Asian climates is crucial.

### 3.1.2   Design of Floaters

Six ongoing projects to reduce basic manufacturing and installation cost for floating offshore wind turbines are listed in table 1. NEDO provided 10 billion JPY for supporting these projects. By leveraging Japan's shipbuilding technologies and infrastructure, such as docks, technologies will be developed to optimize floating bases and mooring systems. Low-cost construction technologies are also being studied to realize the world's first mass production system for floating turbines.



**Table 1.** List of potential projects with floater types and companies

| Floater type | Company |
| --- | --- |
| Semi-submersible | Hitachi Zosen Corporation KAJIMA CORPORATION |
| TLP (Tension Leg Platform) | MODEC, Inc. TOYO CONSTRUCTION CO.,LTD. Furukawa Electric Co., Ltd. JERA Co., Inc. |
| Semi-submersible | Japan Marine United Corporation NIHON SHIPYARD CO.,LTD. "K" Line Wind Service, Ltd. TOA CORPORATION |
| SPAR | TEPCO Renewable Power, Inc. Tokyo Electric Power Company Holdings, Inc., TODA CORPORATION |
| Semi-submersible | Tokyo Gas Co., Ltd. |

Table 1 indicates that the semi-submersible type is predominant in these projects. This is largely because the SPAR, with its large draft, requires a specialized assembly port or a calm sea area with sufficient water depth (100 meters or more), or special technologies for installation. For instance, the 2MW SPAR in the Goto Project was installed using a SPAR-upending method in deep waters near the installation site after being towed out from the port at shallow waters. In contrast, the semi-submersible type can be assembled in much shallower waters. Given Japan's port conditions, where water depths are typically around 20

meters at maximum, the semi-submersible type appears to be more advantageous for installation and is likely to be earlier. Further investigation on efficient installations is needed for larger SPARs with bigger wind turbines.

Optiflow targets reducing the building cost of the float by utilizing light weight structure. The key concept consists of a guy-wire-supported and slanted tower at the center, connected to three columns via slender lower-hulls and via wires to reinforce the strength of the floating structure. Single-point-mooring and turret allow the system to rotate and weathervane around the

mooring point. The barge type targets reducing the building cost by utilizing quay-side installation. Such new float designs may give solutions, but more efforts are necessary towards reducing the cost. But, a collaboration with wind turbine manufacturers is inevitable for better solutions.

Development of vertical-axis wind turbines has been minimal, with the exception of the SKWID project by MODEC, where unfortunately, the substructure supporting the wind turbine sank after water ingress during a severe storm in 2014, in transit

phase (Möllerström et al., 2019). A more recent innovation is found in the FAWT (Floating Axis Wind Turbine) concept developed by Albatross Technology. What sets this system apart is its unique power take-off mechanism, which differs greatly from the conventional gearbox and power generator setup. This design also has the advantage of using simple, straight-shaped blades, which could potentially reduce costs and allow for domestic manufacturing if successful.

## 3.2   Mooring design

The floating offshore installations has been developed in the oil & gas industry, and over 200 FPSO units are already in service(Wang, 2020). Despite academic and industrial efforts, continuous failures of mooring lines have been reported in the Oil and Gas industry. The failure rate reported (Ma et al., 2013; Fontaine et al., 2014a; Spong et al., 2022) is 2.2 to 3.0 x $10^{-3}$ mooring lines / year, which is high considering the consequence of the mooring line failure. The causes of the failures are also





reviewed in 2022 (Spong et al., 2022), fatigue, corrosion and wear are identified as the major causes of failures. Instead, the
failure by the overload is few, meaning the strength design is generally well done in the industry. The challenges of mooring
design and research work is described in the appendix.

Several research studies are being conducted by a Japanese organizations. Kyushu University and Nippon Steel Engineering
has studied the wear prediction of mooring chains(Takeuchi et al., 2021; Gotoh et al., 2019; Ookubo et al., 2022). Japanese
contractor, JMU, is conducting the offshore demonstration with scaled model to understand the design, constructability and
robustness of the chain-fibre rope hybrid mooring under the scheme of NEDO's Green Innovation Fund(JMU).

Japan faces harsh natural environment with complex seabed topography, complicated soil variety, regular tropical cyclones
and earthquakes. Because of the lack of the offshore industry in the past, the soil information around Japan is scarce. Offshore
oil and gas platforms exist in offshore Australia and Gulf of Mexico in US, where tropical cyclones pass, and the regional guide-
lines are available from those countries(ABS, 2011; Australian Petroleum Production and Exploration Association Limited,
2017). However, the oversea guidelines are not Japan specific. Considerable efforts need to be made to resolve the technical
challenges of the mooring design and to establish Japan regional specific design rules and guidance.

## 3.3 Electrical facilities

A floating offshore wind farm is planned for construction at a considerable distance from the shore, which requires a substation
to facilitate high-voltage transmission and minimize power transmission losses. Research by Jump et al. (Jump et al., 2021)
explores the cost-effectiveness of the floating solution, with case studies indicating that the transition water depth, at which
floating substructures become the more economical choice, varies between 55m and 60m for different sites. However, this
transition water depth is site-specific and subject to change, influenced not only by environmental conditions but also by
supply chain dynamics.

The world's first floating substation was implemented in the Fukushima Floating Farm Demonstration Project (Fukushima
Offshore Wind Consortium, 2024). Given the relatively small size of the wind farm, the substation is correspondingly smaller,
featuring a 32MW transformer and a 66kV GIS (Yoshimoto et al., 2013). In contrast, numerous bottom-fixed offshore substa-
tions have been constructed, with topside weights for recent HVAC substations ranging from 1,140t to 4,800t and capacities
varying from 210MW to 400MW (Offshore renewable energy CATAPULT, 2018). Since the topside weight of floating substa-
tions being relatively modest when compared to floating oil rigs, the construction of floating substations is expected to pose no
significant challenges. Various floating substation concepts have been proposed (DNV, 2024; BW Ideol and Hitachi Energy,
2024; Semco Maritime, 2024; Sevan SSP, 2024).

Conversely, the application of a floating solution for High Voltage Direct Current (HVDC) substations presents unique
challenges. The topside weight is substantial, and the valve tower is susceptible to motion-induced stresses from floater motion.
For instance, the bottom-fixed HVDC substation DolWin Kappa (Simens Energy and Dragados Offshore and Tennet, 2024),
designed as the centerpiece of the DolWin 6 wind farm, features an 11,297t topside with dimensions of 31.6m height, 77.5m
length, and 36.5m width, housing large valve towers for a transmission capacity of 900MW. The topside, designed to endure



harsh conditions, necessitates significant motion reduction, making the construction of floating HVDC substations particularly challenging. Several floating concepts for HVDC substations are found (BW Ideol and Hitachi Energy, 2024; Nevesbu, 2024).

In the case of floating solutions, dynamic cables are essential due to the motion of floaters. Array cables in bottom-fixed offshore wind farms typically operate at 33kV or 66kV, export cables from HVAC substations usually have a rating of 220kV (Larsson, 2021). The Fukushima project successfully employed 66kV cables (Fukushima Offshore Wind Consortium, 2024), but the technical maturity of 220kV three-phase dynamic cables for HVAC transmission remains a challenge. The lazy-wave configuration is a widely adopted approach for stress mitigation in dynamic cables. Yang et al. (2021) and Yan et al. (2022) have explored optimization methodologies for dynamic cable configurations. Additionally, Ahmad et al. (2023) studied an optimized configuration for suspended inter-array cables which is suitable for a deep sea. In addition to optimizing the configuration, cable protections, such as dynamic bend stiffeners and touchdown protection sleeves, are employed to mitigate cable stress (Offshore renewable energy CATAPULT, 2021).

The configuration optimization is crucial for designing a reliable cable system, especially for the heavy and large-diameter 220kV AC cables, which pose challenges due to bending radius requirements. In contrast, the design of HVDC export cables, typically rated at 320kV, is comparatively less challenging, as DC cables are single-core. However, cable arrangement becomes a challenge as the substation connects to many interarray cables, requiring careful consideration to avoid contact with mooring lines in harsh environments.

## 3.4 Demonstration projects

As mentioned in the Introduction, the demonstration of profitability and stability is important for a bankable project. Technical uncertainty of floating offshore wind is another challenge for projects to be bankable. Several demonstration projects have been conducted in Japan; namely Goto project, Fukushima-forward project, and Hibiki. Details of these projects are provided in the appendix.

### 3.4.1 Green Innovation Fund

Starting in FY2022, the METI and NEDO started a project to develop and demonstrate elemental technologies for low-cost floating offshore wind turbines, funded by the Green Innovation Fund of the government, which amounts to 2 trillion yen in total. The project aims to establish technologies that will enable power generation costs of 8 to 9 yen/kWh under certain conditions by 2030, and to commercialize floating offshore wind turbines at an internationally competitive cost level. In June 2024, two projects adopting semi-submersible FOWTs were awarded in the "Cost Reductions for Offshore Wind Power Generation" category under NEDO's Green Innovation Fund Phase 2 site leasing program. One project, led by C-Tech, will demonstrate a hybrid semi-submersible FOWT over 15MW off the coast of Aichi Prefecture. The goal is to establish technology that can commercialize floating offshore wind power under specific conditions (e.g., strong wind under tropical cyclones) at internationally competitive prices by fiscal year 2030. The other project, led by Marubeni Offshore Wind Development, will demonstrate two semi-submersible FOWTs over 15MW off the coast of Akita Prefecture, in waters 400 meters deep. These projects commenced in July 2024 and are set to conclude in March 2031, with commercial operations expected to begin in autumn of 2029.





In September 2024, NEDO selected five "next generation floating offshore wind power technology development" projects, including the FAWT project, one TLP FOWT project, one semi-submersible FOWT project, and two spar FOWT projects. Unlike the previously mentioned demonstration projects, these involve a variety of substructure types. The selection of the FAWT project, which utilizes less mature technologies, came as a surprise, as did the inclusion of the TLP project. Despite the spar type's apparent disadvantages due to port water depth, it was selected with the aim of advancing installation technology.

NEDO's support for the offshore wind industry remains strong and continuous.

## 3.5 Operation and Maintenance

### 3.5.1 Learnings from ongoing FOWT projects and research projects

The importance of operation and maintenance and continuous efforts from both research and industry to deliver a variety of solutions and to answer current and future challenges contributes undeniably to the development of a highly promising floating
offshore industry. Bigger turbines and larger and more distant wind farms have required new approaches and technologies (World Forum Offshore Wind). The tow-to-port maintenance has been employed for the heavy maintenance for two well-known floating offshore projects: the Kincardine and the Hywind Scotland. The Kincardine is the first commercial-scale floating offshore wind farm to be operational in 2021. It is situated about 15 km off the coast of Aberdeen, and consists of five WindFloat units with five 9.5 MW turbines and one 2 MW turbine, and all are installed on WindFloat semi-submersible
platforms. During the summer of 2022, one of Kincardine wind turbines suffered a technical failure and a major component required replacement. Thus, the turbine was disconnected, towed to shore, and subject to heavy maintenance. In May 2023, a second Kincardine wind turbine, after a major component failure, encountered the same situation of heavy maintenance and was towed from Scotland to the port of Massvlakte, Rotterdam. These wind turbines were the first cases of floating offshore wind turbines in the world to encounter the complexities of heavy maintenance. Important lessons emerged from the situations
linked with heavy maintenance of the Kincardine floating offshore wind farm: the duration of turbine shutdown, the associated costs, identification of appropriate port for operation and maintenance, and availability of a secure fleet. The first Kincardine turbine encountered the following shutdown duration: 14 days in the port for maintenance, 52 days from disconnection to turbine reconnection, 94 days from turbine disconnection to end of post-reconnection. Lessons such as the distance from the wind farm to the port should be taken into account, and the port should have a deep-water quay and sufficient room. In
addition, the port must be equipped with a heavy crane, and the busy status with other activities should also be taken into account. The total cost for the vessels contracted for maintenance of the first turbine of Kincardine exceeded the amount of 4 million dollars. These high costs emphasized the necessity for floating offshore developers to take into account planning of maintenance contracts and to secure fleet contracts with Anchor Handling Tug Supply (AHTS) through frame agreements in order not to be exposed to very high market rates and market tightening. Strategic maintenance planning and intense research
about alternative maintenance strategies were also among the lessons from Kincardine heavy maintenance (Pacific Northwest Center of Excellence for Clean Energy; North American Clean Energy).





Important lessons for the operation and maintenance of floating offshore wind farms have also been offered by the world's first floating offshore wind farm, Hywind Scotland, which opened in 2017 and is located off the coast of Peterhead, Scotland; It consists of 5 wind turbines of 6 MW each and a SPAR-type foundation. Equinor, the Norwegian operator, announced in 2024

that after more than 6 years in operation, all 5 wind turbines of the Hywind Scotland required a heavy maintenance. Heavy maintenance was needed in a sheltered and controlled environment. Taking into account the absence of required infrastructure and skills in Scotland, all the Hywind Scotland wind turbines were towed to shore, to the west of Norway, by the Wergeland base in Gulen. Wergeland was seen to be the closest port with sufficient water depth and offshore wind experience that could service the Hywind Scotland turbines; it was the same place where the Hywind Tampen was mobilized - see Figure 3. The

heavy maintenance of the five wind turbines was carried out by the Wergeland Group together with heavy lifting and engineering company Sarens and the turbine supplier Siemens Gamesa (European Commission (Technical University of Denmark and Renewable Energy Institute); Equinor, 2024; Energy Voice, 2024). According to an Equinor's statement offered to the AJOT(American Journal of Transportation, 2024), the heavy maintenance consisted mainly in replacement of the main bearings for all 5 wind turbine of Hywind Scotland. As a note, the Hywind Scotland has delivered for many years high-capacity

factors (a capacity factor of 54 %), and has been seen by Equinor as a pilot project which can offer valuable lessons for the future floating offshore wind farms around the world. The replacement of main bearings is perceived also by Equinor and its partners as an important source of learning in terms of improvement of operation and maintenance for the commercial floating offshore wind projects. Moreover, Equinor aims to support initiatives in order to reduce heavy maintenance and to develop efficient onsite repair solutions for future floating offshore farms(American Journal of Transportation, 2024). As per

opinion of some experts in offshore industry, the needs for heavy maintenance for the Hywind Scotland was "unsurprising", but they considered as unusual and intriguing the heavy maintenance requirement for all five wind turbines and the decision to tow to port at once all the 5 units of Hywind Scotland (Energy Voice, 2024). After few months of heavy maintenance, in October 2024, Equinor announced that all five turbines of Hywind Scotland have returned to the offshore site and reconnected (Equinor, 2024). Regarding the tow to port strategy, linked for instance with multi-line anchors, the tow-to-port strategy is

viewed as reactive and unscheduled response to failures. Therefore, future research initiatives and projects need to focus on proactive maintenance strategies, where the tow-to port strategy will be developed into a carefully scheduled and planned event for the maintenance of wind turbine (Dinkla, 2024).

Furthermore, according to the World Forum Offshore Wind (WFO), the tow-to-port heavy maintenance approach is not seen to be a feasible solution for particular commercial-scale floating offshore wind farms, and new solutions for an onsite

maintenance are recommended. onsite heavy maintenance is seen to bring important advantages in terms of reducing downtime and repair time and avoiding the necessity of disconnection of floating offshore wind turbines. As an important note, heavy maintenance providers are implementing their strategies based on the priorities of floating offshore wind farms' stakeholders. Furthermore, cost reduction is paramount for all floating offshore wind farm projects. Predictive maintenance strategies and remote monitoring technologies need to be taken into account for a well-planned heavy maintenance and a spare parts strategy

(World Forum Offshore Wind).





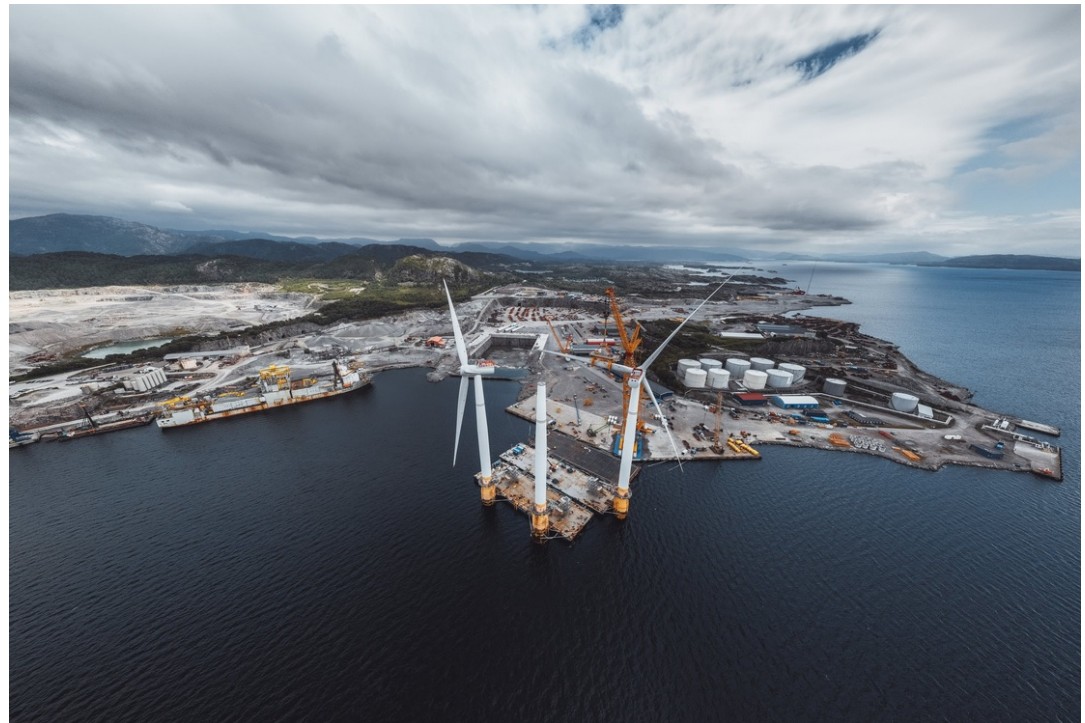

**Figure 3.** Maintenance of floating offshore wind turbines at shore, in Norway (Source: Wergeland (Wer)).

As an innovative in situ operation, in 2024, the Kincardine floating offshore wind farm encountered the first in situ major component exchange for a wind turbine without the need to tow the unit back to a port. The replacement operation of a generator for a 9.5 MW turbine was performed offshore, on site, by making use of a GenHook up-tower crane which was temporally installed on the top of turbine, see Figure 4. The operation was executed from an offshore support vessel (OSV) which was supported by crew transfer vessels (CTVs). The old generator was lifted from the nacelle on the deck of floater, and after transferred on an offshore support vessel (OSV) by making use of a GenHook up-tower crane. The new generator was transferred from the OSV on the floater and lifted on the nacelle. The timeframe for execution of this operation including full mobilisation and demobilisation of equipment covered less than one month. This is the first time for the floating offshore industry, when major component exchanges can be done offshore, and without the usage of massive offshore cranes, or the need to tow to port the wind turbine. The use of up-tower crane technology proved that tow-to-port operations can no longer be required for several types of floating turbine repair. The use of tower crane technology has opened a new chapter in the maintenance of floating offshore wind. (Offshore Channel by N. Hashemi, 2024; OffshoreWind.biz by A. Memija, 2024).

The current implementation and best practices of standards, technical regulations, and conformity assessments in both the Japanese and European offshore wind power markets bring to attention three periodic inspections for the operation and maintenance phase for floating structures: annual inspections (in principal, documents checking); interim inspections (every 2 to 3 years) and periodic inspections (every five years). For both interim and periodic inspections, an inspection plan and in-



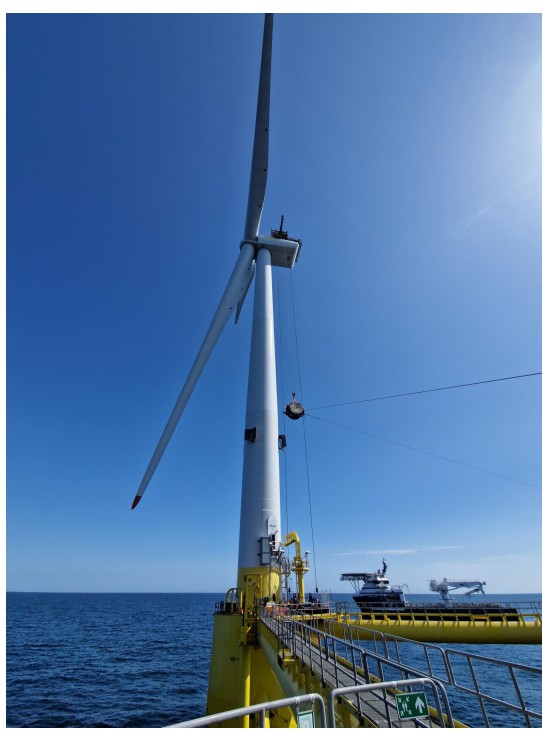

**Figure 4.** Major component exchange with the help of a a GenHook™ up-tower crane, at the Kincardine offshore wind turbine, on site, south of Aberdeen (Source: Offshore Channel (OffshoreWind.biz by A. Memija, 2024)).

spection procedures are well prepared in advance. Moreover, interim and periodic inspections are conducted on the plans and procedures which obtained approval from classification societies. During interim inspection, the visible area is inspected, while during periodic inspection, underwater structures and inside floating structures are inspected (European Commission 590 (Technical University of Denmark and Renewable Energy Institute)).

Floating offshore farms are seen to introduce challenges and constraints from an operation and maintenance (O&M) perspective (McMorland et al., 2022). The lessons and experiences gained in the European and Nordic countries are seen to open up many opportunities for learning in many parts of the world, particularly Japan. An increased distance from shore is associated not only with stronger and constant winds, but also with harsher weather conditions which will impact the operations, 595 reliability and maintainability of FOWT and limit accessibility to sites. A challenging situation is also linked to the majority of the European ports and harbours which are not prepared to deal with scale of operations required for maintenance of FOWT. The Tow-to-Side (T2S) strategy was analyzed by different studies, but key aspects such as time to disconnect, towing speed, and weather limitations are still pending to become standardized. Weather windows are seen as vital for cost and safe operations, but are also associated with an increase in travel time. It was found that one of the main costs associated with operational 600 expenditure (OpEx) is represented by what is called the opportunity cost of downtime which is simply defined as revenue that could have been generated if the wind turbine had been in operation (McMorland et al., 2022). One of the challenges linked





to the operation and maintenance for floating offshore wind is seen to be the cost and complexity linked to mooring lines. The design and integrity of mooring lines are among the critical factors. In this regard, Dinkla (2024) investigated the effectiveness of multiline anchor systems to improve the cost efficiency associated with the operation and maintenance of floating offshore wind farms. The results from simulations and analyzed case studies (Morro Bay California and Gulf of Maine) have recommended a broader adoption of multi-line anchor systems for the floating offshore wind farms. The multi-line anchor systems can contribute to improve reliability and cost-effectiveness, and can lower the costs linked to operation and maintenance and the levelized cost of energy for the floating offshore wind farms. Moreover, the multi-line anchors can have environmental implications, by reducing the footprint on the seabed (Dinkla, 2024).

The costs of O&M for offshore wind turbines can represent more than 35% of the LCOE. Currently, maintenance has been shifting to reliability or predictive condition maintenance with the support of the digital twin (Nejad et al., 2022).

With regards to offshore wind turbines, particularly, large wind turbines greater than 2.5 MW or 3 MW, the condition-based maintenance has been viewed as essential for the drivetrain components such as gearbox, main bearings and generator. Lessons from the first floating offshore wind turbines drew attention to the dynamic behavior and life of drivetrain, particularly, main bearings. The outputs of the condition-monitoring systems for larger offshore wind turbines can be employed to support fault diagnosis and prognostic models such as physics based, data driven, and hybrid models(Nejad et al., 2022). Together with the acoustic emission condition monitoring, SCADA based condition monitoring and vibration based condition monitoring are the most commonly employed types of condition monitoring for the wind turbine drivetrain. However, the vibration based condition monitoring is the most widely condition monitoring, mainly because of an easy instrumentation and a reliable response (Nejad et al., 2022; Randal, 2011). In terms of data, SCADA-based Condition monitoring is making use of data collected at operational wind farms. Within a research and industrial context, SCADA data is considered to be a valuable data which is readily available. However, there are many challenges about valid data which can be used for condition monitoring and how to extract it from a huge amount of recorded data (Nejad et al., 2022). An overview of the state of the art brought to attention that intense research is currently being done on condition monitoring and failure prediction of offshore wind farms using SCADA data, machine learning, and artificial intelligence (Chesterman et al., 2023). SCADA data as a practice are sampled at around 1 Hz and averaged every 10 min, and this practice is regarded as economical since no installation of extra sensors is required (Nejad et al., 2022). Today, the focus of many research studies is on SCADA data with 10 minute resolution, but research is possible on SCADA data with a higher resolution such as 1 minute, 10 seconds, 1 second, and this will allow detection of short-lived failures. This approach is indicated to be done for data which originate from real operational wind farms and preferably, from several operational wind farms. Moreover, it is recommended to make SCADA data public in order to develop standards and improve techniques for the analysis of prediction errors (Chesterman et al., 2023). The quantity and and quality of the SCADA data depends on the turbine manufacturer. In order to analyse the SCADA data, the machine learning techniques are used such as unsupervised or clustering learning and supervised or classification and regression learning. Moreover, there are different types of SCADA monitoring methods such as clustering, trending, damage modeling, assessment of alarms, and expert systems, and normal behavior modeling (NBM) (Nejad et al., 2022). The NBM is a popular methodology for the





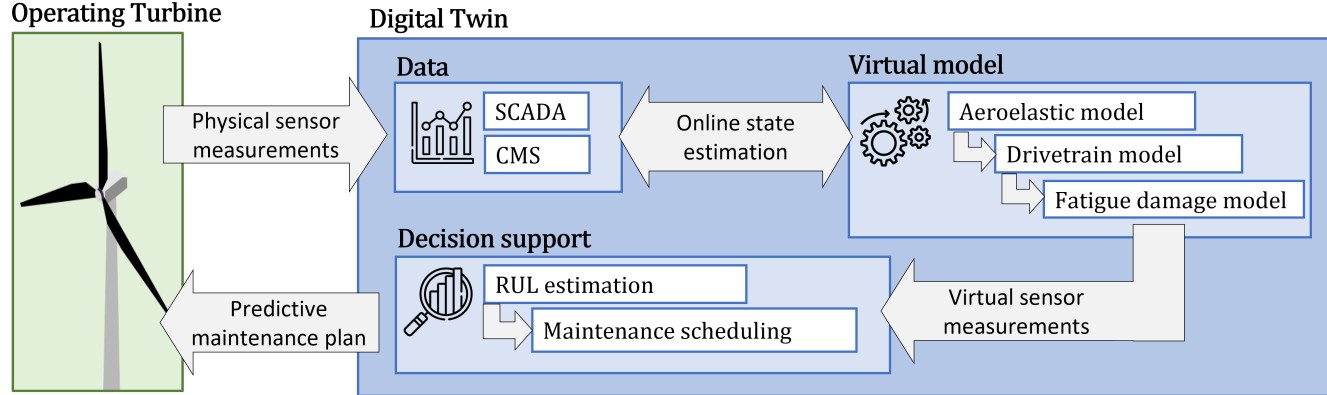

**Figure 5.** Example of a digital Twin framework for continuous RUL estimation in wind turbine drivetrain components (Mehlan et al., 2023).

SCADA data and there are different NBM models such as statistical, traditional machine learning, deep learning (Chesterman et al., 2023).

### 3.5.2  Digital Twins

The usage of digital twins for condition monitoring of floating offshore wind is seen as an emergent and essential research direction (Nejad et al., 2022). Digital twin can offer many benefits for offshore wind farms such as the reduction of operational and maintenance costs, improvement of their market competitiveness, holistic condition monitoring, predictive health monitoring (Ibrion et al., 2019; Mehlan et al., 2023). Predictive maintenance is a part of the condition-based maintenance (CBM) and is aimed to be used in the offshore industry in order to decrease operation and cost related and to increase reliability and availability of offshore wind turbines(Mehlan et al., 2022). Fundamentally, there are key points in terms of building a digital twin: sensors and measurements, model development (data-driven model, physical-based modeling, hybrid model), performance of data analysis, interpretation and validation for fault detection, calculation of remaining useful life and operational decision making (Ibrion et al., 2019). An example of a digital twin framework for a Remaining Useful Life (RUL) in the wind drivetrain components and towards a predictive maintenance plan is offered by Mehlan et al. (2022) and Mehlan et al. (2023), see figure 5.

The digital twin framework shown in the Figure 5 is making usage of the measurements from SCADA and the drivetrain condition monitoring system, and proposed virtual sensors. It is a digital twin employed for the monitoring the accumulated fatigue damage and RUL in the drivetrain bearings and for virtual sensing of the wind turbine aerodynamic hub loads. The physics-based analytical models are used for estimation of local bearing loads and damage and the data-driven regression models are used for aerodynamic load estimations (Mehlan et al., 2023). A digital twin approach can be used for a hybrid testing method for the nacelle testing in order to predict the full load response for the drivetrain in a wind turbine nacelle. This digital twin approach refers to the high-fidelity virtual model of a nacelle and a test bench, model which was validated by making usage of partial load measurements through a physical nacelle testing. This digital twin is used to predict the





load responses beyond testing capacity of a nacelle test bench and can contribute to reduce the costs of conducting expensive nacelle tests (Siddiqui et al., 2023b). In comparison with a scaled testing, the digital twin approach can reduce uncertainties which are linked to a physical scaling. Moreover, the digital twin approach can be employed to solve challenges faced by existing test benches in order to test larger and larger wind turbines. The trend of building bigger nacelle test benches in order to accommodate the needs of larger wind turbines is seen as unsustainable(Siddiqui et al., 2023a).

Location of offshore wind farm far from maintenance ports and shorelines, dynamic motions in floating offshore wind turbines, narrow weather windows and unfavorable weather conditions can seriously impact maintenance planning and optimisation (Jacobsen, 2023). In addition to the condition based maintenance, an effective planning and an optimization of the O&M related vessels can reduce the high costs. Various types of vessels are being used for O&M such as the CTVs (crew transfer vessels), SOV (service operation vessel)s, vessels able to perform heavy lifting, towing to shore, mooring and cable laying vessels, anchor handling tug supply vessels. Subsea components such as mooring lines, anchors and substructures require underwater visual inspections and involvement of remotely operated underwater vehicle (ROV). Moreover, the vessels related to the O&M can vary from each wind farms and can have considerable contribution to increase the O&M related costs (Jacobsen, 2023). On the direction of improvement of logistics and optimization of O&M practices in UK, and particularly, in order to reduce the need for long and many transfers between onshore O&M base ports and floating offshore wind farms, two strategies can be used: one strategy which utilizes SOV for maintenance and another strategy which makes usage of OMB (offshore maintenance base) which accommodates CTVs (Avanessova et al., 20222). A SOV comprises many facilities such as accommodation for staff, an walk-to-work gangway, a maintenance and spare parts platform, a launch and recovery system for a daughter craft. An OMB can have a foundation shared with a substation or can have a totally separate foundation and be connected to substation via a bridge. An OMB and a substation can also share the helicopter base and emergency recovery system (Avanessova et al., 20222). Taking in consideration the costs and energy, SOV is seen as a preferred strategy, but carbon emissions are lower with the usage of OMB. Taking in account, the weather data, the associated costs with vessels can further increase. The costs linked with OBM can decrease, in case the OMS shares foundation with a substation (Avanessova et al., 20222).

Towards the improvement of logistics and optimization of O&M practices for the Norwegian offshore wind, the study of the Jacobsen (2023) looked into new methods to lower costs for the operation and maintenance for offshore wind and how the the OMB can effectively contribute to cost reduction in offshore wind industry. Norway can make usage of its many offshore oil and gas platforms, installations and floating substations that can be used as OMBs. The results of the study of Jacobsen (2023) have shown that OMBs can significantly lower the costs for operation and maintenance for the Norwegian offshore wind for both areas dedicated to bottom fixed and floating development on the Norwegian Continental Shelf. Within the industry, the Norwegian company Fred. Olsen Windcarrier has analysed the feasibility of the OMSs concept, namely, "Windbase" which is based on a modular design. The Fred Olsen Windcarrier supported a study and analysis to be done about three different scenarios linked to the "Windbase" OMS: a scenario which investigated the performance of one SOV equipped with a helipad and accommodation for technicians; a scenario involving 2 SOVs with walk-to-work capabilities alongside the "Windbase" OMS; a scenario which involved 3 CTVs for transfer of technicians. All three scenarios have shown high availability, but





the most efficient performance with 98 % availability rate was offered by the third study/scenario for the "Windbase" OMS (Jacobsen, 2023). However, a major challenge for the implementation of the OMBs with CTVs is represented by the harsh conditions of the North Sea as the CTVs have an operational limit of less the 2.0 metres which is available for only half of the year period as per Copernicus data(Jacobsen, 2023). In 2021, a partnership was signed between Fred. Olsen Ocean together with its subsidiaries Global Wind Service and Fred. Olsen Windcarrier and the Japanese Shimizu Corporation. This partnership brought together knowledge, experience, capabilities, related supply chain, in-depth knowledge of the Japanese market and was seen as beneficial for development of the Japanese and international offshore wind (Fred. Olsen Windcarrier).

As a key recommended area for modeling of O&M for FOWT is to have case studies not only the from European areas, and particularly, the Northern Europe, but also from other areas around the world (McMorland et al., 2022).

Regarding the O&M in Japan, the Japanese companies are trying to integrate within their business experiences and expertise, world leading operation and maintenance technology, know-how and services providers. An illustrative example is offered by the recent agreement signed between the Deutsche Windtechnik and Nippon Steel Engineering (NSE) to jointly offer operation and maintenance services for the offshore wind farms in Japan (Offshore Wind).

Improvement of the maintenance rate is essential to improve the operation rate. According to a report of the Fukushima-forward project, accessibility was an issue, precisely, the scene where the crew transfers onto the wind turbine from/to a working vessel or CTV (Crew Transfer Vehicle). It is reported that the wave height of about 1.5 m was the limit. The large swell on the Pacific Ocean may have contributed a lot to the difficulty of accessibility, and it is necessary to solve this problem specific to Japan by using access gangway and improving its performance. The large number of lightning strikes is also a problem, especially on the Sea of Japan side. Technology to reduce the frequency of maintenance, such as technology to control lightning strikes, will be necessary. Development of drone-based inspection, underwater robot-based inspection, AI-based failure detection/identification technology, remote monitoring technology using digital twin technology, and preventive maintenance should also be seen as necessary and effective ways (Fukushima Offshore Wind Consortium).

## 3.6 Summary and key takeaways

Various design of floaters have been considered in Japan. Design of mooring has challenges from unique metocean and geology conditions. The configuration optimization of electrical facilities connecting substation connects to many inter-array cables require careful consideration to avoid contact with mooring lines in harsh environments. Various demonstration projects are being conducted to reduce uncertainty reduction for predictability.

O&M of floating offshore farms introduce challenges and constraints with longer distance and harsher weather conditions. Japan has additional challenges of swell and lightning strikes that decreases the availability of access for maintenance. Digital twins for condition monitoring is seen as an emergent and essential research direction (Nejad et al., 2022). In addition, adequate development of fleet taking in the experience from the world is important.





# 4 Industry development

## 4.1 Introduction

Developing industry is critical for the successful development and scaling of floating offshore wind. To meet the national target, 200-300 of 10 MW class floating offshore wind turbines will need to be built and installed annually. The technology, infrastructure and the workforce to achieve the mass construction and installation of such FOWT should be developed in Japan. Here we discuss supply chain and human resources which are integral to the growth.

## 4.2 Supply chain

Offshore wind farm is comprised of tens of thousands of parts, including turbine components, floating platforms, mooring systems, and power cables. Japan has related industries all over the country, providing high compatibility and high potential for domestic industrial development for FOWT. The construction industry is strong nationwide, and the shipbuilding industry is very active. In addition, there are strong industrial base for industries that can provide materials for substructures, mooring

lines and anchors, such as steel, concrete and chemical products. The Public-Private Council set a target for the domestic procurement ratio to be 60% by 2040 to promote private investment in the Japanese market, to create a resilient supply chain for stable energy supply, foster industries, and establish cost competitiveness.

### 4.2.1 Fabrication

To realize the mass and rapid production of floating offshore wind turbines in Japan, one of the biggest challenges is where

and how to build the floating body. Depending on the type of floating body and construction method, it is reported that the fabrication of floating body and structures accounts for about 19% of LCOE and more than 50% of construction process(Stehly et al.; Noguchi et al., 2023). For scaling the production to an industry level, it is necessary to design an efficient fabrication flow and develop the infrastructure to support continuous long-term construction. As Díaz and Guedes Soares (2023) pointed out, optimizing the fabrication process and its logistics is important from the perspective of LCOE as well.

For the fabrication of floating offshore wind turbines, various facilities are required such as vast yards, processing facilities, large cranes, docks, etc. Crowle and Thies (2022) summarized that 15 functions are needed for fabrication including substructure component fabrication and assembly, blade construction, storage space for the nacelle and tower, loadout facilities, etc. They suggested that those functions could be achieved by one or a combination of multiple ports. At the stage of the pre-commercial demonstration in Europe, existing shipyards and fit-out quays have been utilized.

Japan has many shipyards and using them for the manufacture of floating bodies is a promising option for efficient fabrication. Because an additional investment is not necessary to build the floating body, manufacturing in shipyards is better for the project from the perspective of CAPEX. However, their location and dock size can be a constraint. For example, the size of the floating body is restricted by the size of the dock. Moreover, most of the shipyards are located in western Japan and far away from promising sea areas like Hokkaido or Tohoku area. This can lead to challenges in planning operations for towing



the floater, as the accuracy of weather forecasts is generally considered to be within 72 hours(Noguchi et al., 2023). Because of this, the shipyard for substructure fabrication should be located within areas that can be reached in such duration.

Instead of dividing the port for the fabrication and assembly of floaters, aggregating all functions into one port and developing a dedicated infrastructure for FOWT can be another option. The fabrication flow can be optimized for the delivery of FOWT and lead to the most rapid and inexpensive process. However, huge investments and space are necessary to develop such integrated ports. For most projects, the initial investment is too large, especially when future market demand is uncertain, and it is also difficult to secure such a large space for the development of an integrated port.

As described so far, how to design the FOWT's fabrication process including where to construct is not obvious, and it is an optimization problem relating to trade-offs of investment cost and performance such as fabrication speed, cost, and so on. The optimal solution depends on the geographical condition and the stage of development. Because infrastructure development takes a long time and requires massive investment, it is important to make a strategic decision and have a long-term plan for general Japan.

Research exploring efficient fabrication flow for mass and rapid production has not been studied as much in Japan. One of the reasons is the unique constraints in Japan, where there are many candidates for shipyards, but the land space of each yard is limited. The research can be separated into two topics, manufacturing processes in shipyards and flow optimization among wind farms, shipyards, and base ports. These topics have been studied separately or without the perspective of fabrication. In terms of improving the fabrication processes in shipyards, there are many papers in the field of shipbuilding. Mitsuyuki et al. (2014) proposed a simulation platform of shipbuilding named pDES, which provides functions to define a model of the construction workflow of a ship, simulate the workflow, and output cost and duration of the workflow. Okubo and Mitsuyuki (2022) applied the simulation to a case manufacturing hull blocks of ships and demonstrated the usefulness of the simulation. Aoyama et al. (1999) proposed a shipyard simulator based on Petri nets. These studies consider resource restrictions on facilities and human resources to support production scheduling. To perform the simulation accurately, historical data about the construction process are essential. Aoyama et al. (2021) developed a shipyard monitoring and visualization platform based on sensors and video data. Shinoda and Tanaka (2016) proposed a method to observe workers' behavior using mobile devices for risk assessment of shipyards. As a study on not only the shipyard but also FOWT, Laura Castro-Santos and Brage (2020) emphasized the production process in the shipyard and developed a cost estimation method that included activity costs. As for the construction of floating bodies, there are not so many achievements in Japan. In conducting demonstration projects, it is important to record data around the fabrication process precisely and make it possible to utilize the data when considering the fabrication process afterward. Research on the locations and logistics of floating offshore wind farm deployment focuses primarily on the site selection and installation process and does not deal with the logistics of the fabrication. Mytilinou and Kolios (2019) proposed a techno-economic optimization method for wind farms based on the life-cycle cost, including the fabrication. Although the paper mainly targeted fixed-bottom offshore wind farm site selection, it provides a general framework to evaluate and optimize wind farm projects. Díaz and Guedes Soares (2023) pointed out that most studies of modeling logistic requirements and installation mainly deal with installation vessel planning and optimization considering weather conditions. They emphasized the importance of construction logistics and conducted an evaluation of wind farms including the some elements of construction





and logistics. When designing supply chains in Japan, it is important to take into account the fabrication process, and for this purpose it might be useful to integrate simulations of the fabrication and data collected from prior demonstration projects which are enabled based on the study on shipbuilding into existing methods of site selection or supply chain logistics.

It is important to explore the efficient fabrication flow to realize mass and rapid production in Japan. However, where and how to construct floating bodies has been less studied. One reason may be that Japan does not have much experience in fabrication, and the research topic will be more attractive after some demonstration projects. Simulation of the fabrication process in shipyards has been established in the context of shipbuilding and is a strong tool for studying the fabrication process of floating bodies. Some research has studied the flow optimization of the supply chain of floating offshore wind turbines from the perspective of transportation and installation. By including the fabrication process in the research scope, it is possible to examine the most efficient supply chain for the mass and rapid production of FOWT.

### 4.2.2 Transportation and Installation (T&I)

The phase of Transportation and Installation (T&I) takes long period and a large cost. Installation costs are projected to account for a large portion of the total lifecycle cost of floating offshore wind systems, approximately 12% to 22% for floating offshore wind systems (Díaz and Guedes Soares, 2023). This is also where significant cost reduction can be achieved through innovative installation concepts, efficient logistics, and learning curves.

T&I of floating offshore wind systems are non-standard as they depend on the size of the rotor, the type of foundation, the development of technology and the conditions of the site (Jiang, 2021). The T&I sequence for various floating structure designs, namely SPAR, semisub, and TLPs, is reviewed in (Chitteth Ramachandran et al., 2022). The installation of superstructure is conducted in different phases depending on the substructure design, making the substructure design the dominating factor of T&I planning. The required infrastructure for T&I, such as port facilities and vessels, is also dependent on the FOWT design. Detailed description and challenges for T&I design, especially on the interconnection of infrastructure and weather conditions, are provided in the appendix.

The current port infrastructure in Japan is insufficient to meet the target of 30 GW in 2030 and 45 GW in 2040, and development plans are being explored. Ports in Japan are relatively small and dispersed compared to the ports in Europe. Considering pre-assembly and storage, bearing load for crawler cranes, and quaywall length for berthing, the area required for a port to handle 50 systems of 10MW were estimated as 22.2ha, with a large portion dedicated for temporary storage of imported parts (Tetsushi Noguchi and Oshima, 2021). The monopiles tend to be larger and heavier in Japan to withstand the seismic force from earthquakes in Japan. In addition, the development of ports are currently considered based on the requirements set for bottom-fixed structures, and a development plan with a long-term perspective including FOWT is necessary.

Simulation tools for modeling T&I are useful during the planning stage of infrastructure design. These system-level simulations provide prediction of the cost and schedule of the T&I phase with higher resolution of the dependencies of tasks and the cost structure. Torres et al. (2023) applied a simulation-based approach with a high level representation for the T&I phase of complex offshore wind projects to considers key activities, operations, and resources needed to complete the build. Discrete-Event Simulation are often deployed to simulate these interdependencies. For example, Barlow et al. (2015) developed a hybrid





framework that combines a discrete-event simulation and robust optimization of the installation schedules against weather un-
certainty. Another model by Díaz and Guedes Soares (2023) estimates the cost for each wind farm concept considering the
supply of resource capacity (technicians, vessels, staff) and the unit times of consumption of resource capacity by products
and services, among others related to the technology, deployed that incorporate the operational details of the duration of the
process and the weather conditions.

These simulation tools rely on weather forecast data or stochastic weather data to assess the effect of weather downtime. The
metocean condition of Japan has two unique characteristics, tropical cyclones and harsh winter conditions. Tropical cyclones
rarely hit the region of interest, but the swells caused by these cyclones can bring long-period waves in the Pacific Ocean,
leading to longer weather downtime. The northwestern wind in winter is associated with strong wind and waves in the Northern
area of Japan. In practice, approximation for operationability has been conducted by a factor called a "service coefficient."
Wada and Ozaki (2014) developed a stochastic model for significant wave models based on the Markov process considering
the monthly variability that can be used for a more accurate estimation with time-domain simulation. The operation considering
weather limitations was presented using an agent-based simulation (Honda et al., 2022).

As mentioned above, the limiting conditions for offshore operation are relatively low in Japan. Under these limitations, the
harsh winter weather in the Japan Sea will sustain the installation process for a large portion of time, leading to large area
requirements for the temporary stock of floating structures (Noguchi et al., 2023). If we need to travel more than 3 days, the
low accuracy of weather prediction will require allocation of safe harbor. Learning from offshore operation in Europe, both in
terms of infrastructure and training of the workers is critical.

Considering the lack of port infrastructure in Japan, we need to explore innovative and inexpensive methods of large-scale
mass construction and installation, including assembly on barges. The idea of floating platforms for assembly and installation
has also been proposed (Noguchi et al., 2023). The bearing loads required for some activities are smaller than for others. By
assigning ports to conduct dedicated activities with lower bearing load requirements, the investment in port infrastructure can
be reduced. Effective inventories management, such as the development of local supply chains, can help to limit the requirement
of land area for ports. Integration of a system to manage the supply chain can also contribute to the reduction in the required
port space. For example, collaborations with automobile manufacturers for a lean supply chain can help.

To facilitate installation at higher wind speeds and with less human intervention, a trend has been observed in the use of
specialized lifting, mating, and damping devices. Current offshore installations are heavily dependent on installation vessels
and skilled technicians. Automation has been a large trend in the construction industry. Although automation of offshore
processes are highly desired, robotics and AI are still in its development phase (Mitchell et al., 2022). Large and powerful
autonomous systems must accurately transfer and install heavy components in challenging weather conditions (Jiang, 2021).
As full autonomy is still a challenge, we need to train the operators to meet the needs of increasingly demanding. Real-time
simulation models are utilized to assess performance under realistic operational conditions for training the operators and to
assess the operation, e.g., the SFI MOVE project (Hong et al., 2024).

The significant challenges in logistics, vessel availability, and port infrastructure create a substantial barrier to large-scale
deployment. The port must have ample storage capacity to provide buffer of supply chain, a robust bearing capacity for the





heavy components, and a wide and deep berths to accommodate the installation vessels. However, the ports in Japan are
relatively small in size. It is crucial to invest in the necessary infrastructures for Transportation and Installation (T&I) to address
these constraints. As can be seen, the performance of T&I strategy is an emergence of an complex system that is interdependent
on design, weather condition and infrastructure. The problem needs to be addressed with an integrated approach.

### 4.2.3 Integrated view of supply chain

In the case of floating offshore wind, the supply chain for floater faces some unique challenges. Large-scale floaters require
specialized fabrication facilities. Due to the size and weight of the floater, transportation and installation (T&I) require careful
planning of the infrastructures. Coordination between various suppliers, manufacturers, and logistics providers is crucial to
ensure efficient operation. Compared with offshore oil and gas platforms, floating offshore wind is unique in the number of
structures that need to be installed. The projects must be executed so that they can supply sufficient numbers of structures and
also achieve economies of scale. The supply chain must be discussed considering the dependency and connection between
fabrication, transportation, assembly, and installation. Since the product being built is large, the capacity for intermediate stock
is often limited and costly to expand. Weather conditions cause irregular suspension of the flow. These conditions require close
coordination, and synchronization among the supply chain is inevitable for an efficient and robust execution. In the analysis
of systems engineering of floating offshore wind systems, deployability and maintainability have been highlighted as criteria
for a cost-competitive design (Barter et al., 2020). Also, a trade-off for substructure designs between compatibility of a design
with a port and a design with operational stability has been pointed out (Barter et al., 2020). As the performance of the supply
chain is an emergent of the complex interaction of various phases of project design, an integrated performance assessment
framework is inevitable.

In Europe, agent-based simulations are used to assess and predict the performance of project execution given the existing
infrastructure (such as work vessels, construction infrastructure, work conditions, and weather data) in the project design phase
(Torres et al., 2023). These simulations can also be utilized to develop work plans with fine granularity, such as staffing in the
project execution phase.

Developing a simulation model that can evaluate the performance of the supply chain is critical, as Japan is in the phase of
infrastructure design. A simulator that can handle manufacturing, transportation & installation in a single model can be used
for optimizing the supply chain. A study has conducted a case study for supply chain and identified the weather limitation as
the primarily bottleneck for FOWTs (Sobashima et al., 2025). Here, knowledge obtained through individual projects can be
stored for knowledge sharing. Knowledge of weather conditions around Japan can also be shared in this platform. The lack of
existing infrastructure provides an opportunity to optimize the supply chain.

In addition, the opportunity for the contractor to start earlier with a new project could be evaluated. Aligning such interests
might result in a change in preference. Finally, considering $CO_2$-emissions as a performance indicator of the considered
strategies (in addition to installation time and costs), would enable a contractor to quantify its contribution to combat climate
change, which may be a competitive advantage during the tendering processes, and to substantiate investments in $CO_2$-
reduction systems. No method for close coordination and synchronization among stakeholders is developed.



There are still gaps in the literature that cover transportation and installation strategies, especially for FOWTs.

### 4.3    Marine related workforce/human resources in offshore wind

The wind energy workforce in Europe is rapidly growing, with projections estimating the workforce will increase from 300,000 to over 500,000 by 2030 (Baltic Wind EU, Bulsky, K). For instance, the UK's offshore wind industry employed around 32,000 people in 2024, with expectations of more than 100,000 jobs by 2030 (The Crown Estate, 2024). Offshore wind jobs, in particular, hold significant potential for future employment. By 2027, over 574,000 technicians will require industry training, with 80% of these technicians concentrated in 10 countries, including Japan (Global Wind Energy Council (GWEC)).

The offshore wind sector offers substantial opportunities for industrial growth, job creation, and economic development. In Norway, the workforce is transitioning from oil and gas to offshore wind, supported by educational programs like those offered at NTNU (NTNU, b). Norway's expertise in oil and gas could give a head start to the growing floating offshore wind industry. Detailed description of the workforce in wind energy is provided in appendix.

Japan holds strong capacities and capabilities in the field of shipping industry and ship building industry. On the other
hand, the capability for offshore development is limited due to the small number of offshore oil and gas fields in the Japan's water. Development of human resources for the future of offshore industry and renewable energy sources like wind power, was seen as a a fundamental strategy for Japan as an island nation and maritime country. On 20 July 2015, in Japan, at the Grand Opening Ceremony for the Marine or Ocean Day (a national holiday in Japan since 1996), the Prime Minister Shinzo Abe has announced the aim of Japan to improve its marine resource development and to increase dramatically its number of
marine engineers from 2000 to at least 10,000 by 2030. On that special occasion, the "Project to Cultivate Marine Pioneers of the Future" was announced, and a consortium made by governmental representatives, industry, research and academia was to follow up and to implement the announced directive. For instance, in 2021, this consortium partnered with more than 28 universities, 55 organizations, 22 companies, 4 national research institutes and one non-profit organization. In this view, the Maritime Bureau of the MLIT has been since working to develop specialized curricula and educational materials
which systematically and comprehensively cover a wide range of knowledge required for ocean development, including and not limited to natural resources, shipbuilding, machinery, and electricity. Through these efforts, the Japanese governmental institutions aim to secure the human resources required by the marine industry by building a human resource development system which will serve as the foundation for the ocean resource development in Japan (Nejad et al., 2019; The Nippon Foundation Ocean Innovation Consortium, 2024). Another important initiative was the establishment of the Nippon Foundation
Ocean Innovation Consortium on the 4 October 2016. Since that time, the Nippon Foundation Ocean Innovation Consortium has acted as a great and inspiring Japanese initiative for supporting the strategic development of offshore industry and to continuously expand awareness about great potential and vital importance of marine energy resources. Moreover, the Nippon Foundation Ocean Innovation Consortium has supported the capacity building for future generations of ocean engineers, both for students and young professionals through various national and international seminars and training tailored to offer attention
to the attractions of offshore industries, to facilitate learning about technology of offshore development and to convey the latest technological trends in offshore industry. Furthermore, the Nippon Foundation Ocean Innovation Consortium has supported





since 2016 many internships in Japanese and international companies and bespoke summer schools at overseas universities, and grants programs in cooperation with organizations in Norway, USA and UK, and Netherlands.

Regarding the training for offshore wind workforce in Japan, in September 2022, the ClassNK (the Japanese shipping classi-
fication society) signed a Memorandum of Understanding (MOU) with Maersk Training A/S (a part of the A.P. Moller Maersk Group, a Danish shipping giant) for training of the offshore wind farm operators and education for alternative fuel ship crews. Moreover, Maersk Training and GiraffeWork (a subsidiary of parent company Daikyo Kenki) have signed a collaboration agreement to open in 2024 in Kawasaki-City a training center for the wind industry in Japan. This centre will offer courses in accordance to the GWO, a Basic Technical Training (BTT), an advanced rescue training course and first aid training (ClassNK,
2024; Global Wind Energy Council (GWEC)).

Currently, in Japan, there is still a shortage of engineers, skilled employees and experts who have the knowledge of ocean development, particularly, related to floating offshore wind. Japan is viewing the development of human resources and marine related workforce with specialized and high technical skills and expertise for offshore development as an important matter for a sustainable offshore market share gaining. Furthermore, towards an efficient workforce transition in Japan, the development
and continuous updates of the educational programs in order to match the needs of workforce in offshore wind energy are required to be implemented (Nejad et al., 2019; The Nippon Foundation Ocean Innovation Consortium, 2024).

An important matter which requires attention is the maritime cabotage or coastal shipping rules in Japan and their impact on the development of offshore wind, and required vessels and its workforce. Basically, as a primary legislation, the Mariners Act and the Act on Ship's Officers and Boat's Operators requires to all mariners and staff on Japanese flagged vessels to
have Japanese nationality. Moreover, as per the Ships Act, only those vessels which are Japanese owned can have permission to engage in domestic maritime transport of passengers and goods. There are some specified exceptions under international agreements, but as a rule, the foreign flagged vessels shall obtain a special exemption permit in order to operate in the Japanese waters. It is important to highlight the Act on Ship's Officers and Boat's Operators is an integral part of a large framework of laws, regulations, guidance, ministers' ordinances which address not only navigation, but also safety, security, environmental
protection. MLIT is in charge for administration of cabotage rules and the Japanese Coast Guard for their enforcement (Watson Farley & Williams).

The offshore wind farms located in the Japan's waters are legally classified as "domestic ports" or "closed ports", and therefore the transfer of crew, turbines, cables and other materials from mainland to offshore wind farms are subject to cabotage rules. The cabotage rules in Japan highly limit the ability of foreign flagged vessels to be involved in installation and operations
and maintenance of offshore wind sites (European Commission (Aquilo Energy GmbH)).

The Japanese maritime cabotage laws apply to all vessels working with offshore wind power facilities, and the vessels with a foreign flag must either be re-flaged in Japan or to obtain a special exemption which allow for a temporary and limited access. The current process for a foreign-flagged vessels to operate lawfully in the Japanese water is very complex, can take many months and encounter significant costs.
Another alternative is re-flagging of ships in Japan, a process which require time (over one year), high costs, and complex matters such equity requirements (owners need to transfer a majority vessel stake to Japanese owners). Moreover, the vessels





must have as crew Japanese nationals and Japanese-licensed mariners/personnel. There are some exceptions for foreign nationals to be accepted as part of the crew for the vessels with a Japanese flags, but specials conditions and requirements are also in place. Another point is there are limited number of globally accredited training programs for offshore wind workforce (Watson
Farley & Williams; European Commission (Aquilo Energy GmbH)).

The Japanese cabotage law and rules can have a negative impact on both the crew of offshore wind fleet and on specialist vessels which needs to be in sufficient number and on schedule and to have timely access to offshore wind sites. The Japan cabotage can bring time delay, increase costs, can have high negative project finance implications and can negatively impact the acceleration for development of offshore wind in Japan (Watson Farley & Williams).

A recent EU commissioned study highlighted that Japan will benefit from regulatory reforms which aim to remove restrictions on the access of foreign vessels access to the offshore wind projects in Japan. It has recommended that cabotage framework in Japan to remove restrictions on cabotage for offshore wind projects, and to clarify the rules for evaluation of applications regarding awarding a special exemption or special permission for the foreign flagged vessels. Moreover, to considerably increase the number of training and course offerings for offshore wind as there is still a limited number of Japanese
nationals which are qualified to operate and crew specialized vessels required for offshore wind sites. In addition, a relaxation of regulations by making usage only of Japanese nationals as seafarers was also recommended. In order to support the development of offshore wind industry in terms of surveying, installation, operation and maintenance in Japan, the cable - laying vessels, offshore wind turbine installation vessels may require to be supplied by international operators and to have non-Japanese staff and mariners on board (European Commission (Aquilo Energy GmbH); Watson Farley & Williams).

In order to address the risk of maritime cabotage on the development of offshore wind industry, in June 2024, a policy package was introduced by the Japan Financial Services Agency (JFSA) which plans to introduce amendments and special measures in order to utilize both the foreign-flagged vessels and mariners and foreign expert staff for construction and operation and maintenance of offshore wind farms in Japan. However, the cabotage reform will be carefully evaluated from national interests and security concerns perspectives in order not to weaken local industry players and to negatively impact the
employment in domestic market (Watson Farley & Williams).

## 4.4 Summary and key takeaways

This section has focused on the execution phase of floating offshore wind systems. First, and for most, the key feature of this phase is to achieve the economics of scale. Manufacture of large structures has been a made-to-order business in many cases. The introduction of manufacturing methods to improve the efficient and lean production process is critical. The large
investment in the vessel and port infrastructure to meet the T&I process defined by the OWF design is another critical decision. Automation technologies are desired to improve the efficiency and safety of the installation process.

In addition, a holistic view of the FOWT project is required. The dependency of T&I process on the foundation design calls for optimization of engineering design and project design. A system engineering approach for an integrated view and close coordination of stakeholders is necessary.





A strategic approach to identify what the required size and capability of the workforce for offshore wind. Develop and update the education program for students and engineers to match the upcoming needs, e.g. from shipbuilding to offshore wind.

## 5    Contribution to S + 3E

### 5.1    Introduction

After the Fukushima disaster and as a sign towards learning from the 2011 dramatic events, the principles of the Japanese

energy policy have been expressed as *S + 3E* where the major concept and practice of safety needs to become primordial in order to harmoniously balance the Energy security, the Economic efficiency and the Environmental protection. The concept of safety is not just for the safety of workers. It implies the safety of sea, national security, cyber security, energy security, and resilience to natural hazards. According to the Agency for the Natural Resources and Energy in METI, the energy policy in Japan has brought to attention the following approaches: 1. The concept of "absolute safety" cannot be associated with the

energy generation and energy sources, and safety can be pursued only in relative terms; 2. A stable supply of energy anywhere and anytime is unrealistic. Therefore, Japan needs to be prepared for potential risk related to the energy supply, and to build a disaster resilient energy system which is able to ensure a stable supply of energy and to increase its self-sufficiency ratio (Ministry of Economy, Trade and Industry (METI) in Japan ., 2022), (Ibrion et al., 2020a).

The value of offshore wind is not limited to the developers commercial interest, but serves the good for many stakehold-

ers.The Japanese government has set ambitious targets for offshore wind and is also preparing target for floating offshore wind. With the anticipated industry's rapid growth, risks related to safety of people, reliable power supply and cyber security are likely to increased. An increased distance from shore, harsher environment, and weather and environmental factors need to be taken in account for the health and safety parameters which are required to be in place for the FOWT and the workers at sea. For example, the dynamic floater motion in the offshore region are seen to make access and egress of workers challeng-

ing(McMorland et al., 2022). For enhancing occupational safety at offshore wind industry, G+ was established as the global health and safety organization bringing together the offshore wind industry to pursue shared goals and outcomes.

### 5.2    National Security with Scale

The national target of 30-45GW by 2040 was established by the Public-Private Council in 2020, with minimal consideration of utilizing the Exclusive Economic Zone (EEZ). Since then, several gigawatts of projects, mostly bottom-fixed offshore wind,

have been initiated in territorial waters, led by the government under the "Act on Promoting the Utilization of Sea Areas for the Development of Marine Renewable Energy Power Generation Facilities," as discussed in Section 2.2. In 2024, discussions on EEZ usage gained momentum, enabling its extension to an area nearly ten times the size of Japan's land territory. Among various resource potential estimates, Mitsubishi Research Institute reports approximately 1,500GW of wind power potential, factoring in business feasibility and sea routes (Institute, 2024). It is predicted that most of them will be developed by using





floating offshore wind turbines due to deep waters. Technologies for EEZ development need also be advanced, including mooring technologies for deep waters, T&I, Power-to-X(P2X), O&M for wind farm deployed in EEZ.

In Europe, artificial islands are planned as offshore bases for O&M in the North Sea, specifically near Denmark and Dogger Bank. These islands are intended to host a range of facilities, including base ports, heliports, accommodation units, energy storage systems, Power-to-X (P2X) systems, and high-voltage direct current (HVDC) transmission systems. The Danish artificial

island is expected to be completed by 2030, with the Dogger Bank counterpart projected for completion by 2035. Given the deep waters of Japan's Exclusive Economic Zone (EEZ), a similar artificial island would likely need to be a floating structure. Yamamoto et al. (2024) examined the feasibility of utilizing very large floating structures as offshore bases for wind farms in Japan's EEZ. They proposed a semi-submersible platform that could serve as a foundation for floating offshore wind turbine substructures, installation, O&M, and P2X activities. However, they highlighted several technological challenges, including

the creation of calm sea areas, the onsite construction of large semi-submersible structures, and mooring.

### 5.3 Power-to-X (P2X) concepts

Although the scale of floating offshore wind can reach more than 2000 GW, the limited capacity of the power grid will restrict the grid access of such massive introduction. For full utilization of the large capacity, we need a different business concept to utilize and monetize the power generated by offshore wind.

P2X means energy transfer from renewable energy to something else: X. P2X technologies are essential for integrating high-penetration offshore wind energy into the grid by mitigating intermittency. Some promising energy storage technologies are Lithium-ion batteries, pumpsed hydro, green hydrogen, thermal energy storage, redox flow systems, lead batteries, sodium-ion batteries, compressed air/gas, supercapacitors, flywheels, liquid air/gas cryogenic, solid mass gravitational, superconducting magnetic, etc. A list of Japan's policies for energy storage is provided in the appendix.

### 5.4 Cyber Security

Considering cyber security, there are several reasons why offshore and wind farms in general are vulnerable to hackers. The approach for cyber security was focused on IT mainly, without having in mind a different approach for operations technology. The old wind farms, including communication systems, were never designed with the "security by design" mindset like the IEC/ISO 62443 standard. Operational technologies like SCADA and their substations for offshore wind farms do need a

different approach for security compared to IT security. Physical security has often not been sufficiently covered in the design, resulting in a poor quality of locks e.g. applied at wind farm cabinets. Vendor's remote access is not always managed properly (segregation of duties). Communication links to the windfarms can be realized by more than one provider without notice. Use of outdated communication protocols without security enhancements. Thus, a holistic approach to ensure cyber security robustness and resilience is inevitable.





## 5.5 Summary and key takeaways

In order to maximize the contribution of floating offshore industry to Japan, we must consider security. Also, a grand design considering how to scale the future industry with Power-to-X systems is also important.

## 6 Discussion: Recommendation for Future Research

This paper is structured to first identify the challenges in the problem space from a review of social needs. To clearly identify the research gaps in realizing floating offshore wind power in Japan, we first analyzed the unique environmental and social characteristics of Japan. Then, we reviewed the challenges and current state of technological development across various phases, including site selection, design, operation, industry development, and the S + 3E framework. Then we covered the state-of-art technology in the solution space that address the problem space.

In this section, we summarize that the realization of floating offshore wind in Japan is contingent upon addressing a range of research gaps that arise from the country's unique environmental and social conditions in 6.1 and 6.2, respectively. These gaps span from the need for improved data collection and modeling techniques to the development of new infrastructure and regulatory frameworks, and transformation of societal systems, is essential. We organize the common directions for addressing these challenges from the perspectives of digital transformation in 6.3, education and research systems in 6.4, and roadmap development in 6.5.

## 6.1 Gaps caused by unique environmental conditions in Japan: Summarize

Japan's unique environmental conditions create several challenges for the development of floating offshore wind, primarily due to its complex geological and metocean characteristics.

Japan's seabed is highly complex, characterized by steep slopes, deep waters, and a high risk of geohazards. This requires a thorough site selection process, where detailed geological surveys and precise design considerations are essential to ensure safety and stability. However, the existing research and data on seabed conditions are insufficient to meet these demands. A significant research gap emerges in developing comprehensive site investigation techniques that can effectively address these unique geological challenges. This gap also affects the ability to optimize foundation and mooring system designs, making it crucial to establish methodologies that are specifically tailored to Japan's unique seabed conditions.

Another major issue is the lack of sufficient metocean data. Offshore wind projects rely heavily on accurate meteorological and oceanographic data to optimize the design and operation of floating turbines. In Japan, the scarcity of long-term metocean data presents a challenge, particularly in dealing with extreme weather conditions like tropical cyclones and extratropical cyclones. Without sufficient data, it becomes difficult to design floating offshore wind turbines that can withstand these conditions. This creates a gap in design and operational planning, where more accurate and region-specific metocean data, combined with advanced modeling techniques, is required to ensure the reliability of the systems in extreme weather scenarios.





Japan's exposure to large swells from the Pacific Ocean and the influence of the Kuroshio Current further complicate the design and operation of offshore wind farms. These factors can affect not only the stability of the floating structures but also the transportation and installation (T&I) process. The research gap here lies in developing design rules and installation strategies that consider these dynamic ocean conditions, particularly in mitigating the risks posed by long-period swells. The ability to forecast these oceanic phenomena more accurately is also critical for ensuring both operational safety and efficiency.

Finally, Japan's unique geographical shape limits access from ports to offshore wind farms, especially in regions where the waters are deeper and farther from the shore. The combination of long distances and challenging sea conditions introduces logistical complexities during both the construction and operational phases. This creates a gap in infrastructure planning, where there is a need to develop efficient transportation, installation, and maintenance strategies tailored to Japan's port infrastructure and its environmental limitations.

## 6.2   Gaps caused by unique social conditions in Japan

In addition to environmental challenges, Japan's unique social and industrial landscape presents several obstacles to the development.

    One key issue is the diversity of stakeholders involved in Japan's coastal areas. These include fisheries, shipping routes, and local communities, all of which have vested interests in the use of maritime resources. The process of site selection must therefore balance a range of competing demands, which complicates the planning and development process. There is currently a lack of government leadership in coordinating this stakeholder engagement, leading to inefficiencies and delays in project approvals. The research gap here involves developing more effective frameworks for stakeholder management and government-led site selection processes that can streamline approvals while balancing the interests of all parties involved.

    Another significant challenge lies in Japan's underdeveloped industrial base for offshore wind. Unlike countries with a well-established oil and gas industry, Japan lacks the necessary infrastructure—such as large ports, heavy-lift cranes, and facilities designed for offshore operations—that is critical for the manufacturing, transportation, and installation of floating offshore wind systems. This gap in industrial capacity affects the entire lifecycle of offshore wind projects, from fabrication to installation and maintenance. Research is needed to develop supply chain strategies that leverage Japan's existing maritime industry, such as shipbuilding, while addressing the infrastructural deficiencies. This includes the development of dedicated ports, storage facilities, and transportation networks that can handle the unique demands of floating offshore wind turbines.

    Furthermore, Japan has a history of lagging behind Europe in terms of regulatory and standards development for offshore wind projects. This is partly due to the absence of major domestic manufacturers that can push for the adoption of cutting-edge technologies and standards. This results in a lack of consistent rules and guidelines that can be applied across different projects. The research gap here lies in developing a unified regulatory framework that can drive innovation and ensure safety while adapting to Japan's specific environmental and industrial context. The framework should also include a more proactive approach to updating standards in line with global best practices, particularly in areas such as foundation design and operational safety.



## 6.3 Integrated System View with Digital Transformation

Integration among phases is important, as technology components are likely to influence multiple phases of the project. One example is the design of floaters, as it not only characterize the performance during operation, but has influence on the fabrication process and T&I process. Several social needs can be met by an improvement of one technology component. An improvement of another technology component to meet a social need may end up in making situation worse in a different social needs. The floater with best performance during power generation may not be the most cost effective as it may require more welding in fabrication, or have narrower weather window for transportation and installation.

This is caused by the complex dependency structure of the socio-technical system of offshore wind. The floating offshore wind system is complex, where components have strong dependency and system level emergence have topological effects. Emergent behaviors of systems arise due to the interactions and dependencies between system components. Understanding these relationships is crucial for designing, analyzing, and managing complex systems. The importance of integrated view of the system is discussed in the discussion. A systemic view of the tradeoff for optimizing the floating offshore wind system is necessary.

Model-based system engineering (MBSE) uses models as the central element of the engineering process, improving communication, traceability, and efficiency throughout the system lifecycle by managing complexities of the systems of interest. The integrated system view can improve design quality by simulating and analyzing models as engineers can identify and address potential issues early in the design phase. We also expect reduction of defects as Model Based Engineering (MBE) tools can automatically generate documentation, test cases, and code, minimizing human error and improving product reliability.

Digitization is also a powerful tool in the operation phase. Maximizing power production, but also monitoring structural integrity is important for efficient operation. This is achieved through a digital twin, which is a digital replica that integrates real-time data from sensors, dynamic numerical simulation models, and other sources to provide insights into the performance, behavior, and condition of the physical entity.

Digital transformation with Model-based Systems Engineering and Digital twins can connect the full spectrum of design phase to operational management. In addition, MBE tools and Digital Twin tools have potential for synergy and reuse. The modeling capabilities are shared in the design phase and operation phase. By carefully developing standards, modular design, and libraries, we can significantly improve efficiency, reduce costs, and accelerate innovation.

## 6.4 Interdisciplinary education and research

Many of the research gaps identified require interdisciplinary research. The offshore wind has become broader and has required a more interdisciplinary approach. However, existing educational programs are not designed to support and encourage interdisciplinary skills. The focus is very narrow for nowadays classical educational programs, and mainly encourages an in-depth approach of a particular field. From one point of view, this can be perceived as a positive side, as students and engineers need to master fundamental matters related to a particular field and become specialized in a particular discipline. However, most of the graduates lack a broader perspective and lack a holistic and interdisciplinary view. Moreover, interdisciplinary indus-





trial collaborations are also very limited. A sustainable ocean industry, particularly, the offshore wind, requires building and development of adaptive and innovative educational methods which need to address the interdisciplinary aspects. Education for offshore wind needs to become a transformative and a participatory process in order to incorporate societal needs, and to support a viable industry and a sustainable development of offshore wind (Nejad, Amir R. and Ibrion, M., 2021; Ibrion and
Nejad, 2021).

Regarding the Japanese national approaches for the education and training of engineering students, and also young engineering professionals with interests in the field of marine resource development, the Nippon Foundation Ocean Innovation Consortium offers an illustrative example. The Nippon Foundation Ocean Innovation Consortium has successfully managed to bring together many Japanese universities, Japanese companies, public institutions and organizations with abroad well known
universities and experienced companies under the platform of the "Ocean Engineering Summer School". As an example, starting from 2017, the Marine Technology (IMT) at the Norwegian University of Science and Technology (NTNU), Trondheim, Norway has successfully hosted the "Ocean Engineering Summer School" and has greatly collaborated towards this excellent initiative (Nejad et al., 2019).

Among the recommended educational approaches at IMT, NTNU, Norway, the direction has been to implement a fair balance
among the fundamentals of each discipline and to employ applied research results as part of teaching educational materials. Team-based learning and Research-based learning have been proved to be among the recommended educational solutions for development of multidisciplinary marine engineering and especially, interdisciplinary offshore wind. Furthermore, the importance of building human resources for offshore wind requires continuous collaborations not only among universities at national and internal levels, but also among research and industry in order to develop educational and research projects which
reflect the industrial needs and requirements (Nejad, Amir R. and Ibrion, M., 2021; Ibrion and Nejad, 2021; Nejad et al., 2019).

The Japanese students and young professionals which attended to the "Ocean Engineering Summer School" at the IMT, NTNU, Trondheim, had the opportunity to use the simulator facilities at NTNU which are located at the Norwegian Maritime Competence Center (NMK) in Ålesund . The simulator facilities include a large variety of simulators which are used for both teaching and research, see Figure 6. The NTNU Ocean Training is owned by NTNU Ålesund and delivers maritime courses
and training to officers and crew for offshore and merchant fleet (NTNU, a).

As emphasized by Nejad et al. (2022) in order to address various challenges related to the technologies linked to the floating offshore wind, the interdisciplinary research and collaboration is encouraged to be supported among academia and industry. When it comes to matters related to ocean-based industry, and, particularly, to the offshore wind, all science disciplines are connected, thus, a need to interdisciplinary approach in education, research and industry is urged to be implemented Nejad,
Amir R. and Ibrion, M. (2021); Ibrion and Nejad (2021); Nejad et al. (2019).

## 6.5    Need for Industry Roadmap

This paper has identified various technology gaps to facilitate the sustainable development of floating offshore wind. Addressing these gaps is critical to achieve the most desirable future of floating offshore wind. However, the amount of available



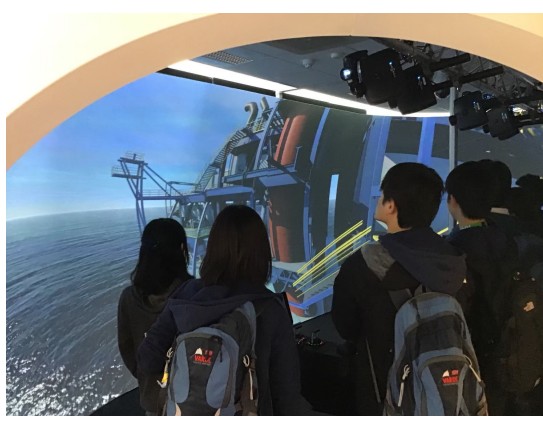
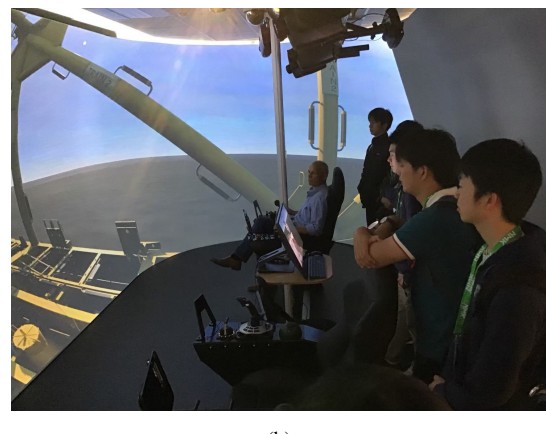

(a) (b)

**Figure 6.** Ocean Engineering Summer School at NTNU simulator (photos by NTNU).

investment is usually limited. Thus, setting a target considering the tradeoff between desirable outcome and feasible investment
shall be explored.

Exploring the tension between desirability and feasibility is critical in designing a project. An approach known as Project
Design (Moser and Wood, 2015) considers the sociotechnical aspect of systems to recognize the interplay of system architec-
ture and project architecture and explore effective strategies for concurrency, phasing and risk management. De Weck (2022)
proposed a structured framework called "ATRA framework" for organizations to strategically manage technology development
and integration to achieve long-term success and maintain competitive advantage. The framework includes the identification
of needs and objectives, technology assessment, roadmap development, technology forecasting, risk management, stakeholder
involvement, and review. Feasibility of investment shall consider the relation between investment in one technology compo-
nent and its impact on the overall project performance. The model-based approach allows quantitative treatment of technology
forecast and its systemic emergence, providing measures for exploration of various roadmaps .

The strategy to address the technology gaps for floating offshore wind shall be designed in the same way for two reasons. One
is the complexity of the socio-technical system, where site selection, system design, operation, supply chain, and the business
model are strongly interconnected. The emerging system performance is dependent on other components, and the systemic
effect is non-negligible. The second reason is the alignment of investment decisions of various stakeholders. The shared target
of floating offshore wind backed up by a sound technology roadmap is critical for the stakeholders to take risks and encourage
investments. The long investment lead time requires "concerted actions" that are implemented in a consistent and timely
manner among the many stakeholders. A technological roadmap that can harmoniously combine industrial and governmental
approaches and initiatives by assessment of readiness of technology, lessons from industry, challenges, opportunities, and
possible technology innovations is critical.





# 7    Conclusions

Floating offshore wind is essential for Japan to tackle the challenges of climate change and energy security. However, continuous efforts for innovation are required to achieve its sustainable development. Reducing the unit cost of energy will be the primary goal, which is driven by higher performance with rational site selection, improved design, and efficient operation, taking advantage of economies of scale by well designed supply chain. The floating offshore wind has enormous potential in Japan, but it needs to be continuously supported by tailored educational and research programs on both national and in-

ternational levels. This paper discussed the challenges and state-of-art research in these fields with a thorough review of the unique situation in Japan. The findings of this paper is summarized in 7. This review identified the remaining gaps that need to be addressed in future research activities. Along with the research opportunities defined in each technical field, the need for integration, digitization, and transdisciplinary research has been emphasized as a common need for the future development of floating offshore wind energy. The need for technology roadmap for floating offshore wind has also been discussed. The float-

ing offshore industry in Japan seeks to develop, but a tight collaboration among governmental institutions and organizations, education, research and industry is inevitable.

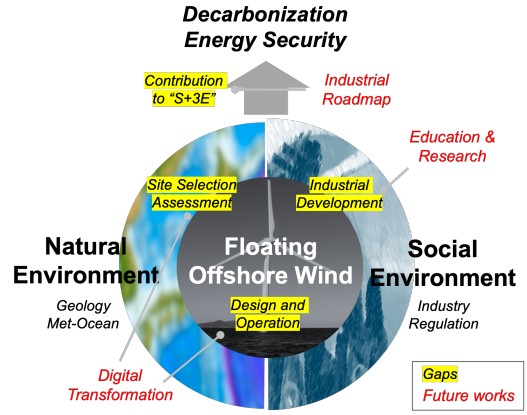

**Figure 7.** Intregrated view required for the sustainable development of floating offshore wind turbines.

## Appendix A:    Grid Investigation, Stakeholder identification, Project Zoning Support

*Grid Investigation*: The availability of the grid is confirmed by the developers by checking the existing grid lines around the project field, the future grid development plan, and the existing power plants around the field. The information of the

existing grid and future development plans is shared by the owner of the grid of each area and OCCTO (Organization for Cross-Regional Coordination of Transmission Operators) according to the guidance from the Ministry and ANRE (Agency for Natural Resources and Energy) (ANR). OCCTO set the long-term grid reinforcement master plan in 2023 to further promote the introduction of renewable energy, and showed their scenario with investments of 6 to 7,000 billion Japanese yen to aim for





carbon neutrality in 2050(OCC). Developers also consider the projection of existing power plant in the project area, especially

their decommissioning plan of the existing plant. There remains uncertainty in the future of nuclear power plants, especially after the tragic tsunami incident in 2011. Even if the grid is occupied by the existing plant, OCCTO may accept non-firm connection subject to the curtailment when the grid is occupied by the other sources. By conducting these advance consultations, the initial developers know the point and location of the grid connection and plan the onshore substation and cable routing for their development planning, which may provide a competitive advantage to the initial developers.

*Stakeholder identification*: A major challenge for ocean utilization is the stakeholder engagement. Various stakeholders are present around the project site. Kimiaki Yasuda and Nagai (2015) showed the process of consensus building in Niigata prefecture and failures in Aomori and Ishikawa prefecture from 2010 to 2015 by introducing 4 main stakeholder categories, i) fisheries relations, ii) prefecture and city halls, iii) local citizens and chamber of commerce, and iv) other official bodies such as education, tourism, environmental activists and port authorities (Kimiaki Yasuda and Nagai, 2015). The developer meets with local stakeholders, especially fisheries cooperatives, to receive recognition and acceptance of offshore wind development as their stakeholder management activities. Currently, it is the norm that each developer conducts the separate and duplicated surveys at the same field. However, the central survey approach lead by governmental body is is being discussed at the time of the writing of this paper to make the bidding process more cost efficient and lower risk for the developers. Compared to the fixed bottom project, the identification and control of stakeholders in the fisheries industry is more complex for floating offshore wind projects due to the fishing right and permission system in Japan. The coastal fishing right is given by the prefectural governor and the offshore fishing permission is given by the Minister of Agriculture, Forestry, and Fisheries. In the current permission system, offshore fishing is not governed by the prefecture, while field zoning is under the prefecture's scope. Fisheries groups outside of the prefecture may exist as stakeholders under the permission of the Minister; therefore, the identification of the stakeholders is difficult for both developers and public offices.

*Project Zoning Support*: Developers often start with project zoning with the associated departments in the prefecture and city halls. The initial project zone is identified considering the field restriction of wind resources, water depth, soil, shipping route, natural reserves, and visual impacts. These project zones are narrowed down by the subsequent hearing of local stakeholders. Considering the buildable area, further study of the wind turbine layout and farm capacity is conducted. Those results are often shared with local stakeholders to promote the project development, with the developer's presentation about the ripple effects and the contribution to the local economy by the offshore wind project. An important concept and implementation framework for ocean utilization is the Marine Spatial Plan (MSP). MSP is being studied and introduced internationally, and IOC-UNESCO (Intergovernmental Oceanographic Commission of the United Nations Educational, Scientific and Cultural Organization (UNESCO)) has published a manual with the aim of clarifying the procedure for introducing the plan. In the Basic Ocean Policy published in April 2023 and decided by the Cabinet, MSP is included to promote the use of multiple marine areas appropriately and effectively, especially for offshore wind. In addition, to centralize marine data, "Umishiru" is to be used and strengthen the functions. The University of Tokyo Ocean Alliance published a guideline for a consensus-building process for marine use to introduce MSP. Accurate and up-to-date spatial data on marine ecosystems, currents, seabed





conditions, and other factors are essential for informed MSP. Ensuring data availability and sharing among stakeholders can be a logistical challenge.

## Appendix B: Demonstration projects in Japan

### B0.1  Goto Project

Japan is one of the pioneering countries in the demo installation of floating offshore wind turbines. In 2013, the first full-scale floating offshore wind turbine in Japan was constructed and installed off the coast of Goto islands, Nagasaki. MoE undertook the Goto project in 2010-2015, outsourcing Toda Corporation as one of the representing companies. The Goto project was the third offshore wind turbine project in the world, following the 2.3 MW SPAR demo by Statoil (now renamed as Equinor), off Stavanger, Norway (2009) and the 2 MW semi-submersible demo by WindFloat, off the coast of Portugal (2011). A 2MW down-wind turbine was mounted and operated on a hybrid-type spar structure with a concrete substructure and a steel superstructure in the Goto project. The project has since led to a commercial project, aiming for a wind farm of a capacity of 16.8 MW with 8 spar units starting in 2026, which will be the first commercial wind farm project in Japan using floating offshore wind turbines.

### B0.2  Fukushima-forward Project

After the Great East Japan Earthquake in March 2011, the national and local governments of Japan have been working together to realize the "Fukushima Innovation Coast Concept" for the reconstruction of Fukushima, and with this as background, a 10-year (2011-2015, 2016-2021) floating offshore wind turbine project, or "Fukushima-forward" project was undertaken. A total of 60 billion yen was spent for the project. The project was expected to create a new energy-related industries. The outcomes of the project are summarized in reports are publicly available. In this project, a wind farm consisting of a 2 MW semi-submersible, 5 MW spar, and 7 MW submersible demo units, as well as a sub-station for offshore transformers, were deployed and demonstrated. After the project, these structures were removed and dismantled.

### B0.3  Hibiki

In 2014, METI and New Energy and Industrial Technology Development Organization (NEDO) started a demonstration project for a floating offshore wind power generation system using a barge-type floating structure that enables quay-side installation with the aim of achieving lower costs. IDEOL is collaborating on this project as a consortium partner. A 3 MW wind turbine started its operation in 2019 in the demonstration area off the coast of Kita-Kyushu. The floating structure targets water depths of 50m-100m. The demonstration is continuing as of writing.





## Appendix C: Challenges of mooring design

About the mooring design of floating offshore wind, various design guidance and standards are published from the Classification societies(ABS, 2020b, a; ClassNK, 2021a; Bureau Veritas, 2019; DNV, 2021a, b), and the standardization organization such as API and ISO(for Standardization, 2013; American Petroleum Institute, 2018). The mooring analysis software are also available such as Orcaflex by Orcina, DeepC/Sesam by DNV and Ariane by BV, but not limited to(Ma et al., 2019). In order to estimate the mooring load, it is imperative to consider the floater motion; therefore, various simulation codes are developed(Barooni et al., 2022) and a number of the comparison study among the codes are conducted with various articles published under IEA Wind Task 30. From Japan, UT-Wind was developed as a numerical simulation by Suzuki et al. (2013). In addition, to evaluate the hydrodynamic load on the Spar structure, SparDyn was created and integrated with MSC Adams and AeroDyn from NREL(Utsunomiya et al., 2014). The performance of the dynamic analysis was validated during the severe tropical cyclone event in Nagasaki Prefecture off the coast of Japan(Utsunomiya et al., 2013).

Receiving the failure report from the industry, various researches have been conducted to the mooring line fatigues, corrosion, and wear. As the result, 2 phenomena are added for the design consideration to the industrial standard API 2SK 4th edition in 2018(Shu et al., 2018) that are Vortex Induced Motion and Out of Plane Bending (OPB) / In Plane Bending (IPB) fatigue. Regarding the Vortex-Induced Motions of floating offshore wind, a state of the art review paper was issued by Fujarra et al. (2012) and by Yin et al. (2022). The latest review paperYin et al. (2022) insists the effectiveness and the limitation of CFD for VIM analysis and the challenges of CFD application to the Semi-submersibles due to the structural complexity comparing to the Spar. In Japan, Rodolfo, Suzuki et al. conducted the experimental research using scaled OC4 semisubmersible model and studied VIM amplitude ratio with respect to the current reduced velocity(Gonçalves et al., 2021). About OPB/IPB, a Joint Industry Project was conducted from 2007 to 2015 (Rampi et al., 2015) after the Girassol incident at offshore Angola(Jean et al., 2005), which reported the multiple mooring lines failure in 8 months after the operation started. Although the JIP provides certain implication of the OPB/IPB phenomena to the fatigue design, and Bureau Veritas issued the guidance after the JIP(Bureau Veritas, 2014), the debates are still ongoing about the practical design method. The wear and Corrosion are also studied as a part of the SCORCH (Sea Water Corrosion of Ropes and CHain) JIP(Jayasinghe et al., 2018). During the course of the SCORCH JIP, microbiological influenced corrosion was found as common issues on the offshore mooring chain(Fontaine et al., 2014b) and DEEPSTAR CTR12402 JIP was formed(Witt et al., 2016). These JIPs insist the difficulty of precise wear prediction. None the less, it is clear that the severe wear occurs in the interlink area with relative chain motions between the links, and the contact area to hard soils such as the touch down zone. As the result of these academic and joint industrial efforts, wear and corrosion margins are revised more conservatively on the API RP2SK 4th revision. From Japanese Organization, Kyushu University and Nippon Steel Engineering has studied the wear prediction of mooring chains(Takeuchi et al., 2021; Gotoh et al., 2019; Ookubo et al., 2022).

The usage of fibre rope is actively discussed due to the potential cost reduction measures of the floating offshore wind as it is widely used in the Oil and Gas industry. In contrast to the over 1,000 m water depth of the Oil and Gas floating platform, the floating offshore wind is generally installed at a shallow water depth of 100-300 m. Polyester fibre rope is commonly used





in the oil and gas industry and Polyester is still prospective material to the floating offshore wind as studied by Utsunomiya
et al. (2016). Due to the number of mooring lines, maintenance difficulties, and shallower water depth in the floating offshore
wind farm, the usage of other fibre rope materials are discussed and a comprehensive review paper was issued in 2015 by
Weller et al. (2015). On the paper, in order to avoid re-tensioning during the wind farm service life, fibre rope materials with
better creep characteristics than Polyester i.e. HMPE and Aramid are considered. However, the long term material property
with visco-plastic and visco-elastic behavior are not yet fully understood for these alternative materials. Japanese contractor,
JMU, is conducting the offshore demonstration with scaled model to understand the design, constructability and robustness of
the chain-fibre rope hybrid mooring under the scheme of NEDO's Green Innovation Fund(JMU).

## Appendix D: Design of T&I

Several port facilities are required for FOWT. An integration port is usually located at the vicinity of the wind farm and used for
installation of the wind turbine on the substructure prior to deployment offshore. The components of the subsystem, imported
or manufactured domestically, are gathered at the port for assembly and integration. A steel substructure assembly port, which
can be further away from the project sites, is an intermediate facility that is used to build steel substructures before being
transported to an integration site. The port is utilized during all offshore wind project development stages, playing a central role
in the catering specialized vessels for site surveys, transportation, installation, operations & maintenance, and decommissions.
The port's infrastructure and inventory space play a critical role in the system integration of the floating offshore wind supply
chain as they function as an interface between land based and offshore activities. The port needs to accommodate the large
fleet of vessels, stock and handle large component for floating offshore wind, host new manufacturing centers, and facilitate
assembly activities. A significant area of land is required, with reinforced quays, enhanced deep-sea harbours and other civil
works to deploy at scale. Heavy-lift vessels, jack-up vessels, barge vessels, and feeders, are required for transportation and
heavy-lift cranes are necessary either at the port or offshore.

Various vessels are utilized during the T&I of FOWT, such as CTV(Crew Transfer Vessel), Tugs, AHV(Anchor Handling
Vessel), AHTS (Anchor Handling Supply Vessel), AHTV(Anchor Handling Tug Vessel), CLV(Cable Lay Vessels), OCV (Off-
shore Construction Vessel), Heavy Lift Cargo Vessel, and SOV (Service Operation Vessel). The specific requirements for the
vessels during each phase of T&I differ among substructure types. The specifications of these vessels are characterized by
features such as deck area, bollard pull capacity, engine power, positioning system, and deck equipment. For example, most
FOWT is assumed to require large bollard pull of 200-300t for towing and hook-up. A study shows the increase in the future
demand for various vessels required the deployment of FOWT in the Celtic Sea by estimating the projected FOWT capacity
and the vessel time required per FOWT unit (Porteous, 2023). When vessels with the required specification are not available,
the operation needs to be conducted by combining vessels. Investment in vessels that meet the quality and quantity of future
FOWT development is crucial.

The vessels and port infrastructure required for the FOWT T&I will have high dependencies on the project design. Although
many bottom-fixed projects have been conducted, the industry has limited experience to identify good practice for FOWTs as





projects move far offshore, harsher environments, and larger scale (Barlow et al., 2015). A research by (Jiang, 2021) provides a state-of-the-art review on various installation methods and concepts for bottom-fixed and floating offshore wind turbines for wind turbine foundations and components, with four visions for the future: 1) vessels that can handle larger blades and
1355 versatility for various support structures, 2) specialized equipment for less human intervention and lower risk, 3) multiphysics simulation tools for installation methods and concepts, and 4) foundations that are installation friendly. At this point, the optimum solution for the installation of floating foundations are still under development. The LEANWIND project (2013-2017) examined logistical challenges related to the deployment, installation, and operation of various foundation solutions, both bottom-fixed and floating, offshore wind turbines. The project explored supply chain optimization for cost reduction with
1360 improved port infrastructure, innovative approaches to vessel design, installation methods, and operational and maintenance strategies.

The performance of T&I is also dependent on the metocean conditions, which are site-specific. There are weather limitations for offshore operations to be conducted. For example, a SPAR type floater, such as HyWind Tampen demonstrated in Norway, requires a calm weather condition (e.g. signficant wave height below 0.5m) (Barlow et al., 2015). As Japan has limited expe-
1365 rience in offshore construction, the weather limit condition are lower compared to the weather limitations considered in UK (Trust, 2015).

Uncertainty in technology development also poses a challenge in infrastructure investment. Larger blade sizes have been installed and the trend is expected to continue (Shields et al., 2021). Shields et al. (2021) projects a significant reduction in the number of turbines as the capacity of each increases. Some studies indicate that the possibility of increasing the cost from such
as T&I may potentially increase the cost, as the large blades will require larger infrastructure investment (Tetsushi Noguchi and Oshima, 2021).

## Appendix E:  Wind energy workforce overseas

The landscape of wind energy workforce is dynamic and can be seen as a hope beacon for many European job seekers and graduates. According to estimations, by 2030, the current wind energy workforce in Europe will boost from 300,000 to more
than 500,000 people. The jobs in the offshore wind are attractive and have a future with great potential (Baltic Wind EU, Bulsky, K). As per the Global Wind Workforce outlook, from 2023 until 2027, the number of new wind technicians is expected to increase by 48,800 on average per year. Furthermore, the number of wind technicians which will require industry training will increase to more than 574,000 by 2027. In 2022, almost 145,000 technicians held at least one valid certificate from the Global Wind Organization (GWO) Basic Safety Training (BST) standard. By 2027, almost 430,000 technicians will require
wind industry training and more than 80% of these technicians will be required in 10 countries worldwide, and Japan is one of them. Moreover, in order to meet energy growth, by 2027, the number of technicians which will require training in operation and maintenance, commerce and industry is expected to pass more than 5100 people in Japan, and from this number more than 1750 will be for offshore wind (Global Wind Energy Council (GWEC)).





Offshore wind industry has a great potential to unveil industrial growth plans to generate employment, to create new jobs,
to support development of supply chain and to boost regional and national economies. An illustrative example is offered by
the case of UK and its very ambitious offshore wind targets. As per the 2024 Offshore Wind Industrial Growth Plan in UK,
in April 2024, the UK offshore industry has employed around 32,000 people and the employment is seen to rise to more than
100,000 people by 2030 (The Crown Estate, 2024).

In Norway, the workforce is shifting from the oil & gas industry to the offshore wind industry and special courses and
training in offshore wind are offered within the framework of special educational programs, one of its kind is offered by the
NTNU, Trondheim, Norway. These kind of programs are supported by the Norwegian Directorate for Higher Education and
Skills. The employees which wishes to transition from oil and gas industry to the offshore wind have different opportunities in
terms of training, education and research in Norway (NTNU, b; Nejad et al., 2019). The workforce and expertise in the oil and
gas industry has the potential to give a head start to floating offshore industry as the process of competence transfer is under
way. The floating offshore industry can play a key role in restructuring the Norwegian offshore industry.

According to a report issued by the Menon Economics and its gross figures, the Norwegian floating offshore wind can
become one the most important job creators by 2050. The floating offshore wind linked to the Norwegian based industry
can support up to 36,000 jobs by 2050, and this depends on the competitiveness and market development of the Norwegian
industry. Furthermore, the Norwegian floating offshore wind could also provide GDP contributions which can reach up 78
billion NOK by 2050. All regions of Norway will benefit from the employment and GDP contributions. The Menon report
pointed out that in terms of value chain, there will be large effects in the maritime industry which has an important role
for development of specialized vessels and port infrastructure. In addition, to the value chain will contribute also specialized
suppliers and sub-suppliers which currently are operating in the offshore industry and various industrial activities. The Menon
economics' scenario have suggested the Norwegian offshore wind market (both floating and bottom-fixed) could support more
than 60,000 jobs in Norway by 2050. As per analysis, the Norwegian offshore wind industry can achieve an annual revenue
by 2050 which can reach up to 115 billion NOK. Definitively, this assumes a successful development of the value chain for
the floating offshore wind in Norway, and taking in account a successful transfer of offshore and maritime expertise. Without
doubt, the Norwegian offshore wind industry can become a critical part of a transition from an economy heavily dominated by
oil and gas. In this regard, sustainable political choices and major commitments and predictable policies will be imperative for
supporting the large potential for employment in offshore wind, and particularly, floating offshore wind (Norwegian Offshore
Wind; GCE Ocean Technology; Menon Economics).

**Appendix F: Energy storage policies in Japan**

Japanese government established the below policies to promote energy storage in Japan.

– Setting Target Prices (2020): To reduce the price and promote the energy storage systems, target prices for commercial
and industrial energy storage systems were defined.Subsidies implemented by the government is only applicable for the
products below the target price for cost reduction.



– Green Growth Strategy(2021): Battery storage is defined as a "new energy infrastructure" that is necessary for the development of green and digital technologies. To promote the wide use of battery storage systems, further cost reduction and promoting reuse and recycling are included.

– 6th Basic Energy Plan(2021): Promoting domestic and industrial power storage systems as technologies necessary for increase of renewable energy. The government will clarify the definition of grid-scale energy storage systems under the Electricity Business Act., and improve the supply-demand adjustment market.

– Energy Storage Industry Strategy(2022): Japanese government defined energy storage systems as the key technology to achieving carbon neutral by 2050 and the essential infrastructure to support digital society. The goal is to establish a
manufacturing base for liquid-based LiB, secure competitiveness in global market, and create the next-generation energy storage market.

– GX Basic Policy(2022): Accelerate green transformation (GX) initiatives to ensure stable energy supply and decarbonization. The goal is to create new demand and markets in Japan, strengthen the competitiveness of the Japanese economy, and lead to economic growth. Energy storage should be one of the important issues and promote investments
to establish manufacturing capacity in Japan.

– Subsidy projects: Support for the introduction of domestic and industrial energy storage systems. Support for the introduction of grid-scale energy storage systems.

*Author contributions.* This paper was initiated by RW, ARN, KI and JS. RW coordinated and led the article writing; creation of the structure; writing of the abstract, introduction and conclusion; and gathering and editing of inputs from all listed co-authors. Site Selection was led by
JS with input from HN and YM. Design and Operation was co-led by KI and ARN with input from YM and KT. Industry development was led by WS, MI, and RW. Contribution to S + 3E was led by JS. Discussion was co-led by TN and RW. The final edit was carried out by RW and ARN together.

*Competing interests.* The contact author has declared that neither they nor their co-authors have any competing interests.

*Acknowledgements.* The second and fifth authors would like to acknowledge the INSPIRE project funded by the "Sustainable Blue Economy
Partnership", a Horizon Europe co-funded partnership, in collaboration with the Norwegian Research Council (project no. 351718). "Nippon Foundation Ocean Innovation Consortium" support is also greatly appreciated.



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
