# Peer review of "Floating Offshore Wind in Japan: Addressing the Challenges, Efforts, and Research gaps"

_Wind Energy Science, 2025_

## Referee Comment (RC1)

- This manuscript presents a comprehensive review of the environmental, social, and technological challenges surrounding the deployment of floating offshore wind (FOW) in Japan. It covers a wide range of topics—from site selection and metocean conditions to design, operation, and maintenance—and draws on international case studies to contextualize Japan's situation. The paper provides useful background and may serve as a reference for early-stage researchers or policymakers seeking an overview of Japan's FOW landscape.

- However, while the breadth of the review is notable, the manuscript would benefit from a significant revision to improve focus, originality, and structural clarity. Below are my detailed comments:

1. Lack of Prioritization Despite Explicit Research Question

   The authors explicitly include the question "What are the gaps and what shall be prioritized?" among their three research questions (Section 1.3). While the paper identifies a wide range of technical and institutional gaps, it falls short of addressing the "prioritization" aspect. There is no framework, criteria, or comparative discussion provided to help readers understand which issues are most urgent, impactful, or feasible. A structured prioritization matrix would strengthen the paper's contribution significantly.

2. Limited Insight on Research Directions for Environmental Constraints

   The review offers a thorough description of Japan's unique geological and metocean challenges—such as complex seabed topography, high geohazard risk, insufficient metocean data, and port-access limitations. However, it largely reiterates the existence of these challenges without proposing how future research might address them.

   Especially, Section 6.1 identifies two major gaps: (1) limited prior research and (2) lack of site-specific data. Yet it does not extend to suggesting potential directions to close these gaps. For example, no specific methodologies are proposed for enhancing site investigations, improving metocean forecasting in cyclone-prone areas, or adapting installation strategies to Japan's long-period swell conditions and port limitations. As a review paper, offering even preliminary suggestions—such as promising modeling techniques, remote sensing tools, or infrastructure planning frameworks—would help the reader better understand how these environmental constraints might be systematically addressed in future work. Including such insights would strengthen the paper's contribution to shaping Japan-specific floating offshore wind research.

3. Need for Explicit Contextualization of International Case Studies

   While the manuscript introduces a range of international case studies on floating offshore wind development—particularly in Europe—these are often presented descriptively and left for the reader to interpret their relevance to Japan. In several instances (e.g., the discussion of tow-to-port maintenance in Kincardine and Hywind Scotland, or digital twin applications for predictive maintenance), there are implicit connections to Japan's environmental or infrastructural conditions, and Japanese examples are occasionally mentioned alongside. However, these links remain largely implicit rather than analytically articulated.

   To strengthen the paper's contribution as a review intended to inform Japan's FOW deployment, it is recommended that the authors move beyond implicit juxtaposition and provide explicit interpretation and contextualization. For example, when discussing European O&M strategies or data infrastructure challenges, what specific lessons are applicable to Japan, and what modifications or local considerations would be required? Highlighting such comparative insights more clearly—perhaps through brief analytical co

4. Minor improvement suggestions

   4.1. Overly Lengthy and Redundant Sections

      The manuscript is excessively long, and several sections could be significantly condensed without loss

of content. For instance:

- The Operation & Maintenance section devotes more than four pages to describing the Hywind Scotland and Kincardine projects. While these are valuable cases, the level of detail provided (e.g., port names, tow duration, exact crane types) is excessive and not directly linked back to Japan's context.

- The Site selection and metocean challenges (e.g., typhoons, earthquakes, steep bathymetry) are described repeatedly across Sections 2, 3, and 4 without synthesis or cross-referencing.

- Similarly, the emphasis on Japan's lack of oil & gas infrastructure and limited metocean data is mentioned in multiple sections with nearly identical wording.

Reducing such repetition and consolidating related content would improve readability and focus.

**4.2. Section 5 lacks clear linkage to the paper's research objectives and Japanese context**

Section 5 is titled in a way that suggests it will examine how floating offshore wind contributes to Japan's societal, environmental, economic, and energy (S+3E) goals. However, this linkage is only briefly mentioned in Section 5.1 (Introduction), and not meaningfully carried through in the subsequent subsections. For example, Section 5.2 (National Security with Scale) introduces European "energy island" projects, but does not explain why these are relevant to Japan or how they relate to national security in the Japanese context. Similarly, Section 5.3 (Power-to-X (P2X) concepts) explains Power-to-X (P2X) technologies in general terms, without specifying how these technologies contribute to S+3E goals in Japan or why they are particularly important in this setting. Section 5.4 on cyber security suffers from the same lack of contextualization.

Overall, the section reads as a collection of general technical topics rather than a focused analysis. It does not provide a systematic or evidence-based discussion of how floating offshore wind supports Japan's S+3E goals, nor does it directly contribute to answering the research question posed in the introduction. Given the already substantial length of the manuscript, I recommend removing Section 5 unless its structure and content are significantly revised to clearly support the core objectives of the paper.

**4.3. Structural Suggestion for Section 3.4**

Section 3.4 contains only a single sub-section (3.4.1). Given that no additional sub-sections are provided, the hierarchical structure appears unnecessarily complex. I recommend removing the sub-section numbering and simplifying the structure to improve readability.

---

## Author Comment (AC1)

**Rebuttal Letter**

**Reviewer #1,**

We thank the reviewer for their thoughtful and detailed comments, which we believe will significantly strengthen our manuscript. The reviewer's insights have helped us recognize that our initial draft, while broad in scope, lacked the necessary prioritization, analytical depth, and structural clarity to effectively address our research questions. We agree that the paper's contribution will be enhanced by moving beyond a descriptive summary to a more focused, analytical discussion that explicitly contextualizes international case studies and proposes concrete research directions relevant to Japan's floating offshore wind (FOW) landscape. We have revised the manuscript extensively to address each of the specific points raised.

**R1-1: Lack of Prioritization Despite Explicit Research Question**

The authors explicitly include the question "What are the gaps and what shall be prioritized?" among their three research questions (Section 1.3). While the paper identifies a wide range of technical and institutional gaps, it falls short of addressing the "prioritization" aspect. There is no framework, criteria, or comparative discussion provided to help readers understand which issues are most urgent, impactful, or feasible. A structured prioritization matrix would strengthen the paper's contribution significantly.

Thank you for your suggestion. We agree that we need to add a structured framework to discuss prioritization. We created a subsection in the Discussion that introduces a framework for prioritizing the identified gaps and the technologies to overcome the gaps.

To systematically address the numerous challenges facing Japan's floating offshore wind sector, a structured framework is proposed to prioritize the necessary research and technological development. This framework first classifies all identified gaps and their corresponding solutions into two primary categories. "Enabling technologies" are defined as those that are absolutely essential for a project's viability; without them, a floating offshore wind farm cannot be successfully built or operated under the required technical, economic, and regulatory performance levels. All other solutions are classified as "Supporting technologies," which, while valuable, are not strictly indispensable for a project to proceed, as alternatives may exist.

The framework then further evaluates "Supporting technologies" against two critical metrics to determine their strategic importance: Cost Reduction and Scalability. Cost Reduction is the primary driver for achieving commercial viability, and activities are prioritized based on their potential to lower the Levelized Cost of Energy (LCOE) by reducing Capital Expenditures (CAPEX) or Operational Expenditures (OPEX), or by increasing Annual Energy Production

(AEP). Scalability refers to a technology's ability to support the mass deployment required to meet national energy targets, focusing on aspects like mass production, supply chain development, and logistical efficiency. This two-metric evaluation helps to distinguish between technologies that offer incremental improvements and those that are true game-changers for the industry. These classifications are synthesized into a prioritization matrix that guides strategic investment and R&D efforts.

"Enabling technologies" are categorized as non-negotiable "Must Have" activities that require immediate attention. Government-led strategic site surveys and scaling up workforce development fall in this category. For "Supporting technologies," those with high impact on both Cost Reduction and Scalability are identified as the "Holy Grail"—the top priorities for long-term research. Development of advanced survey methods and digital twins are in this category. Activities with a high impact on one metric but not the other are considered "Quick Wins," valuable for near-term projects or solving specific issues. This matrix, combined with an assessment of each technology's current readiness level (TRL), provides a clear, strategic roadmap for stakeholders to focus resources on the most critical and impactful solutions.

**Modification: Add section 6.3 Prioritization of Action**

**R1-2: Limited Insight on Research Directions for Environmental Constraints**

The review offers a thorough description of Japan's unique geological and metocean challenges—such as complex seabed topography, high geohazard risk, insufficient metocean data, and port-access limitations. However, it largely reiterates the existence of these challenges without proposing how future research might address them. Especially, Section 6.1 identifies two major gaps: (1) limited prior research and (2) lack of site-specific data. Yet it does not extend to suggesting potential directions to close these gaps. For example, no specific methodologies are proposed for enhancing site investigations, improving metocean forecasting in cyclone-prone areas, or adapting installation strategies to Japan's long-period swell conditions and port limitations. As a review paper, offering even preliminary suggestions—such as promising modeling techniques, remote sensing tools, or infrastructure planning frameworks—would help the reader better understand how these environmental constraints might be systematically addressed in future work. Including such insights would strengthen the paper's contribution to shaping Japan-specific floating offshore wind research.

Thank you for your suggestion. We agree that merely identifying challenges is insufficient and that a review paper's value lies in suggesting solutions We have added a proposal on specific, actionable research directions.

We have revised the **summary for each section** and **Section 6.1** to go beyond simply stating the problems. For each key challenge—e.g., complex seabed topography, high geohazard risk, limited metocean data—**propose concrete research directions as follows.**

**Examples added:**

- **Site investigations:** We suggest specific methodologies like the use of AUVs for high-resolution bathymetric surveys, advanced geophysical techniques, data sharing scheme among other surveys conducted (such as CCS, methane hydrates), application of machine learning to predict geohazard hotspots based on historical data.
- **Metocean forecasting:** Mention the need for dedicated modeling efforts to improve typhoon and cyclone forecasting, integrating satellite remote sensing data, or adapting existing global models for Japan's specific conditions.
- **Installation strategies:** Suggest research into the development of port infrastructure planning models that optimize for limited-access ports.

**Modification**

- Revised the summary for each section
- Revised Section 6.1

**R1-3: Need for Explicit Contextualization of International Case Studies**

While the manuscript introduces a range of international case studies on floating offshore wind development—particularly in Europe—these are often presented descriptively and left for the reader to interpret their relevance to Japan. In several instances (e.g., the discussion of tow-to-port maintenance in Kincardine and Hywind Scotland, or digital twin applications for predictive maintenance), there are implicit connections to Japan's environmental or infrastructural conditions, and Japanese examples are occasionally mentioned alongside. However, these links remain largely implicit rather than analytically articulated. To strengthen the paper's contribution as a review intended to inform Japan's FOW deployment, it is recommended that the authors move beyond implicit juxtaposition and provide explicit interpretation and contextualization. For example, when discussing European O&M strategies or data infrastructure challenges, what specific lessons are applicable to Japan, and what modifications or local considerations would be required? Highlighting such comparative insights more clearly—perhaps through brief analytical co

Thank you for your suggestion. Regarding learning from the Kincardine and Hywind Scotland offshore wind farms, it is important to note that these projects are the first cases of floating offshore wind turbines in the world to encounter the complexities of heavy maintenance and

the evaluation of choices such as tow-to-shore or on site repair. They offer unique opportunities of learning which is very important for FWT development in Japan considering the scarcity of FOWT maintenance experience around the world. As an example of analytical articulation/explicit interpretation and contextualization, the tow-to-port maintenance has been employed for the heavy maintenance for the Kincardine and the Hywind Scotland. These experiences have shown that the default strategy for heavy maintenance—towing the entire turbine to port—is prohibitively expensive, and downtime lasting over three months for a single turbine. For a nascent market like Japan with limited specialized port infrastructure, relying on this reactive and costly tow-to-port model may present a significant threat to project viability.

Modification: Revised Section 4.7

**R1-4: Overly Lengthy and Redundant Sections**

The manuscript is excessively long, and several sections could be significantly condensed without loss of content. For instance:

- The Operation & Maintenance section devotes more than four pages to describing the Hywind Scotland and Kincardine projects. While these are valuable cases, the level of detail provided (e.g., port names, tow duration, exact crane types) is excessive and not directly linked back to Japan's context.
- The Site selection and metocean challenges (e.g., typhoons, earthquakes, steep bathymetry) are described repeatedly across Sections 2, 3, and 4 without synthesis or cross-referencing.
- Similarly, the emphasis on Japan's lack of oil & gas infrastructure and limited metocean data is mentioned in multiple sections with nearly identical wording.

Reducing such repetition and consolidating related content would improve readability and focus.

Thank you for your valid suggestion. We agree that we need to improve the manuscript's readability and focus. We have significantly restructured the paper while reducing and consolidating the contents. At the same time, the paper also aims to provide information to non-experts in the field. Some of what seems to be redundant information to experts can be important to understand the context.

**Modification:** We have fully restructured the paper, so that the content to flow more logically, with less overlap between sections. In addition, we have shortened some of the detailed descriptive passages.

**R1-5: Section 5 lacks clear linkage to the paper's research objectives and Japanese context**

Section 5 is titled in a way that suggests it will examine how floating offshore wind contributes to Japan's societal, environmental, economic, and energy (S+3E) goals. However, this linkage is only briefly mentioned in Section 5.1 (Introduction), and not meaningfully carried through in the subsequent subsections. For example, Section 5.2 (National Security with Scale) introduces European "energy island" projects, but does not explain why these are relevant to Japan or how they relate to national security in the Japanese context. Similarly, Section 5.3 (Power-to-X (P2X) concepts) explains Power-to-X (P2X) technologies in general terms, without specifying how these technologies contribute to S+3E goals in Japan or why they are particularly important in this setting. Section 5.4 on cyber security suffers from the same lack of contextualization.

Overall, the section reads as a collection of general technical topics rather than a focused analysis. It does not provide a systematic or evidence-based discussion of how floating offshore wind supports Japan's S+3E goals, nor does it directly contribute to answering the research question posed in the introduction. Given the already substantial length of the manuscript, I recommend removing Section 5 unless its structure and content are significantly revised to clearly support the core objectives of the paper.

Thank you for your suggestion. We agree that Section 5 was not well focused to effectively link to the paper's core objectives. As to clarify the intention of this section we have restructured the paper and set the focused as a subsection "From Project to National Infrastructure" in the section "Industry and Economic Enablement". It explains the "The challenge is no longer just the operational safety of a single wind farm project, but the comprehensive security of a critical national energy and industrial ecosystem. This expansion of the systemic boundary introduces new and complex challenges that require a holistic grand design."

**Modification:** Condensed the section to a subsection "From Project to National Infrastructure" in the restructured section "Industry and Economic Enablement"

**R1-6: Structural Suggestion for Section 3.4**

Section 3.4 contains only a single sub-section (3.4.1). Given that no additional sub-sections are provided, the hierarchical structure appears unnecessarily complex. I recommend removing the sub-section numbering and simplifying the structure to improve readability

Thank you for your suggestion. We agree that we should modify the structure to improve readability. We have modified the manuscript as follows.

**Modification:** Eliminate the numbering for subsection 3.4.1 and integrate the content directly under the heading of 3.4.

---

## Author Comment (AC2)

**Reviewer #2,**

This paper tries to review the unique challenges of floating offshore wind development in Japan towards the large-scale commercialization, which is very important topic. The paper structure needs to be revised slightly, because the earthquake and the Kuroshio Current, which are mentioned in the abstract and conclusion as the uniqueness of Japan, do not seem to be mentioned enough in the main text.

We thank the reviewer for their thoughtful and detailed comments, which we believe will significantly strengthen our manuscript. The reviewer's insights have helped us recognize that our initial draft lacked the necessary discussion on the uniqueness of Japan's situation. We have revised the manuscript extensively to address each of the specific points raised.

**R2-1 [Title]**

The current title may be misinterpreted as a review of research on floating offshore wind energy system in Japan, but there are few reviews of research papers obtained from the past demonstration projects, and the paper is intended to review what needs to be done towards large-scale commercialization. Just for example, like "Floating Offshore Wind in Japan: Addressing the Challenges, Efforts and Research Gaps towards large-scale commercialization.", some words expressing this aspect are better to be added.

We agree with the reviewer's suggestion to make the title more precise. The current title could indeed be misinterpreted. We have revised the title to better reflect the paper's core theme, which is a forward-looking analysis of challenges and research needs for large-scale commercialization in Japan. The reviewer's suggested title, "Floating Offshore Wind in Japan: Addressing the Challenges, Efforts and Research Gaps towards large-scale commercialization" is a good model, and we decided to adopt it.

**Modification:** Change title to "Floating Offshore Wind in Japan: Addressing the Challenges, Efforts and Research Gaps towards large-scale commercialization"

**R2-2 [Section 2.6]**

This section of metocean conditions should mention about the current, because the authors refer to Kuroshio in the summary section of line1085. The current is also significantly important especially for mooring line design (Also for O&M).

We thank the reviewer for highlighting the importance of ocean currents, especially in the context of mooring design and O&M. We agree that this is a significant metocean factor, particularly given the reference to the Kuroshio Current. We revised Section 2.6 to include a specific discussion on the role of currents, their impact on project design, and the relevance of Japan's specific oceanographic conditions.

Modification: Added subsection 2.2.5 Current Data

**R2-3. [Section 2.6.1 line 306]**

...stochastic simulations such as MASCTO are widely used in Japan "to consider the effect of typhoon", might be more clear what the authors want to say.

We appreciate the reviewer's suggestion for a clearer explanation. We rephrased the sentence to explicitly state that stochastic simulations like MASCOT are used "to consider the effect of typhoons on extreme sea states and turbine loads." This will make the purpose of these simulations more understandable to the reader.

**Modification:** In 2.2.3 rephrased as "Regarding the estimation of extreme wind speed, stochastic simulations such as MASCOT Offshore (Yamaguchi and Ishihara, 2010; Ishihara and Yamaguchi, 2015) are widely used in Japan to consider the effect of typhoon."

**R2-4. [Section 2.6.3 line 343-345]**

The following paper is the outcome from Fukushima FORWARD project about this issue.

A. Yamaguchi, T. Ishihara, Numerical prediction of Normal and Extreme Waves at Fukushima Offshore Site, Journal of Physics: Conference Series 1037 (4), 042022, 2018.

Thank you for providing this relevant reference. The paper by Yamaguchi and Ishihara is an excellent example of work from the Fukushima FORWARD project related to wave prediction. We added this reference to this section to support our discussion on normal and extreme waves at the Fukushima offshore site.

**Modification:** In 2.2.4 rephrased as "In the Fukushima FORWARD project, a wave hindcasting was performed using Wave Watch III, showing good agreement with in-situ measurement data both in normal and extreme conditions (Yamaguchi and Ishihara, 2018)."

**R2-5 [Section 3.1.2]**

Many research in the demonstration projects in Japan or Norway had big efforts to develop dynamic analysis tool to design the optimum floaters. Just as one example, the latest outcome from Fukushima FORAWRD is as follows. Also there are another papers from Goto project, and Demonstration Project of Next-Generation Floating Offshore Wind Turbine. It might be better to refer these latest efforts relating with design of floaters to address the challenge and efforts in Japan or Norway.

T. Ishihara, Y. Liu, Dynamic response analysis of a semi-submersible floating wind turbine in combined wave and current conditions using advanced hydrodynamic models, Energies, 13(21), 5820, 2020

A. Yamaguchi, S. Danupon, T. Ishihara, Numerical prediction of tower loading of floating offshore wind turbine considering effects of wind and wave, Energies, 15, 1-18, 2022.

We agree that the development of dynamic analysis tools for floater design is a crucial effort that deserves specific mention. The provided references from the Fukushima FORWARD project (Ishihara and Liu, 2020; Yamaguchi, Danupon, and Ishihara, 2022) are highly relevant. We revised Section 3.1.2 to include a more detailed discussion of these specific research efforts and reference the provided papers to highlight the progress made in this area.

**Modification:** In 3.1.2 added suggested references.

**R2-6. [Section 3.2]**

The anchor design also is better to be mentioned. "Anchor design against liquefaction due to the earthquake" is one of characteristic of Japan. In Abstract, the authors mention earthquake in line 4, but there is no specific description about the earthquake in the manuscript. Japan

guideline mentions about this liquefaction issue.

https://www.mlit.go.jp/common/001331376.pdf

We thank the reviewer for pointing out this critical omission. The issue of anchor design against liquefaction due to earthquakes is a uniquely significant challenge for floating offshore wind in Japan. We revised Section 3.2 to include a specific discussion on this topic, referencing the provided MLIT guideline to support our claims. This will better connect the manuscript's content with the earthquake reference in the abstract.

**Modification:** In 2.2.1 added "The risk of liquefaction on the anchor is mentioned in MLIT(Ministry of Land, Infrastructure, Transport and Tourism (MLIT), Maritime Bureau, 2023)."

**R2-7. [Section 3.4]**

This section only has Section 3.4.1. Please check the chapter structure.

We apologize for the formatting oversight in the chapter structure. We have corrected the numbering to ensure proper chapter structure throughout the document.

**Modification:** Eliminate the numbering for subsection 3.4.1 and integrate the content directly under the heading of 3.4.

**R2-8. **(Section 4.2.2)**

Line 830: The following paper discuss the swell effect on weather downtime in Japan.

Y.KIKUCHI, T.ISHIHARA: Assessment of weather downtime for the construction of offshore wind farm by using wind and wave simulations, Journal of Physics: Conference Series 753(9) pp.1-11, 2016.

We appreciate the reviewer providing this relevant reference. The paper by Kikuchi and Ishihara is a valuable resource for discussing the swell effect on weather downtime in Japan. We will add this reference to Section 2.2.6 and Section 4.4 to strengthen our analysis of operational challenges.

**Modification:** Add suggested reference to Section 2.2.6 and Section 4.4.

**R2-9. **[Figure 7, Line 1211]**

The uniqueness of Japan found in this paper is difficult to understand from Figure 7. Please consider, if possible.

We thank the reviewer for this constructive feedback. We recognize that Figure 7 may not fully convey the unique aspects of Japan's challenges. We reviewed the figure and its caption to either revise the figure to better illustrate the uniqueness or, alternatively, to provide a more detailed and clearer explanation in the text that accompanies the figure, better highlighting how the data presented relates to Japan's specific conditions.

**Modification:** Prepared a new Figure 7.

**R2-10. [Project names]**

I think that the demonstration project names in Appendix B are not accurate. Please check them. B0.2 →Fukushima Floating Offshore Wind Farm Demonstration Project (Fukushima FORWARD) B0.3→Demonstration Project of Next-Generation Floating Offshore Wind Turbine (The name of floater is Hibiki, but the project name is different.)

We thank the reviewer for pointing out these inaccuracies.

**Modification:** We corrected the project names in Appendix B to the official "Fukushima Floating Offshore Wind Farm Demonstration Project (Fukushima FORWARD)" and "Demonstration Project of Next-Generation Floating Offshore Wind Turbine."

([Noticed typo]Line 596: Tow-to-Side  $\rightarrow$  Tow-to-Shore ? / Line 1211: is summarized in 7  $\rightarrow$  is summarized in Figure 7.)

We thank the reviewer for pointing out these mistakes.

**Modification:** We corrected "Tow-to-Side" to "Tow-to-Port" and "is summarized in 7" has been changed to "is summarized in Figure 7".